# Oligomerization-primed coiled-coil domain interaction with Ubc13 confers processivity to TRAF6 ubiquitin ligase activity

Lin Hu[1], Jiafeng Xu[1], Xiaomei Xie[1], Yiwen Zhou[1], Panfeng Tao[1], Haidong Li[1], Xu Han[1], Chong Wang[1], Jian Liu[2], Pinglong Xu 📷 [1], Dante Neculai[3] & Zongping Xia 📷 [1]

Ubiquitin ligase TRAF6, together with ubiquitin-conjugating enzyme Ubc13/Uev1, catalyzes processive assembly of unanchored K63-linked polyubiquitin chains for TAK1 activation in the IL-1R/TLR pathways. However, what domain and how it functions to enable TRAF6's processivity are largely uncharacterized. Here, we find TRAF6 coiled-coil (CC) domain is crucial to enable its processivity. The CC domain mediates TRAF6 oligomerization to ensure efficient long polyubiquitin chain assembly. Mutating or deleting the CC domain impairs TRAF6 oligomerization and processive polyubiquitin chain assembly. Fusion of the CC domain to the E3 ubiquitin ligase CHIP/STUB1 renders the latter capable of NF-κB activation. Moreover, the CC domain, after oligomerization, interacts with Ubc13/Ub~Ubc13, which further contributes to TRAF6 processivity. Point mutations within the CC domain that weaken TRAF6 interaction with Ubc13/Ub~Ubc13 diminish TRAF6 processivity. Our results reveal that the CC oligomerization primes its interaction with Ubc13/Ub~Ubc13 to confer processivity to TRAF6 ubiquitin ligase activity.

---

[1] Life Sciences Institute and Innovation Center for Cell Signaling Network, Zhejiang University, Hangzhou, Zhejiang 310027, China. [2] Department of Surgical Oncology, First Affiliated Hospital, School of Medicine, Zhejiang University, Hangzhou, Zhejiang 310027, China. [3] College of Medicine, Zhejiang University, Hangzhou, Zhejiang 310027, China. Correspondence and requests for materials should be addressed to Z.X. (email: xia2010@zju.edu.cn)

Protein ubiquitination impacts nearly all aspects of eukaryotic cell biology[1]. This process first involves the enzyme E1-driven activation of ubiquitin (Ub) forming Ub~E1 and its subsequent transfer to a ubiquitin-conjugating enzyme (E2) forming Ub~E2. Ub~E2 then coordinates with a cognate Ub ligase (E3) to ubiquitinate a protein substrate. Repeating of this cascade can result in subsequent ubiquitination of Ub to form a polyubiquitin chain (polyUb)[2]. PolyUb chains linked through Lysine 48 (K48 polyUb chains) in general mark a protein substrate for degradation, whereas K63 polyUb chains are best known for their signaling roles in DNA damage repair, innate immunity, and inflammation[3, 4].

TRAF6 is a prototype of RING domain-containing E3 that has been best characterized for its essential E3 activities in the interleukin-1 receptor and Toll-like receptor (IL-1R/TLR) signaling pathways[5, 6], where it works with the heterodimeric E2 Ubc13/Uev1A to catalyze polyUb chain synthesis. The resulting chains, which could be either conjugated to TRAF6[5, 7] or unanchored derived by direct synthesis[8–10] or by possible en bloc shedding from conjugated TRAF6 by deubiquitinating enzymes (DUBs) with Poh1-like activity[11], activate TAK1, a heterotrimeric kinase complex consisted of TAK1 kinase subunit and TAB1 and TAB2 (or TAB3) structural subunits[12]. Activated TAK1 on the one hand phosphorylates IKK, a kinase complex composed of kinase subunits IKKα and IKKβ and a structural subunit IKKγ, leading to IKK activation[13]. IKK phosphorylates IκBs causing their ubiquitination and degradation, which eventually results in activation of transcription factors NF-κB[14]. IKK also activates the ERK1/2 pathway through a Tpl2-MEK1/2-ERK1/2 cascade[15]. On the other hand, activated TAK1 phosphorylates and activates the p38 and JNK pathways[16]. Together, the activated NF-κB and three MAPK signaling pathways induce expression of downstream genes such as TNF-α, IL-1β, IL-6, and A20 to initiate immune and inflammatory responses[17].

The synthesis of polyUb chains by TRAF6-Ubc13/Uev1A E3-E2 system is highly processive to ensure efficient extension of polyUb chains[18]. Although this processive polyUb synthesis is critical for timely activation of TAK1 in response to

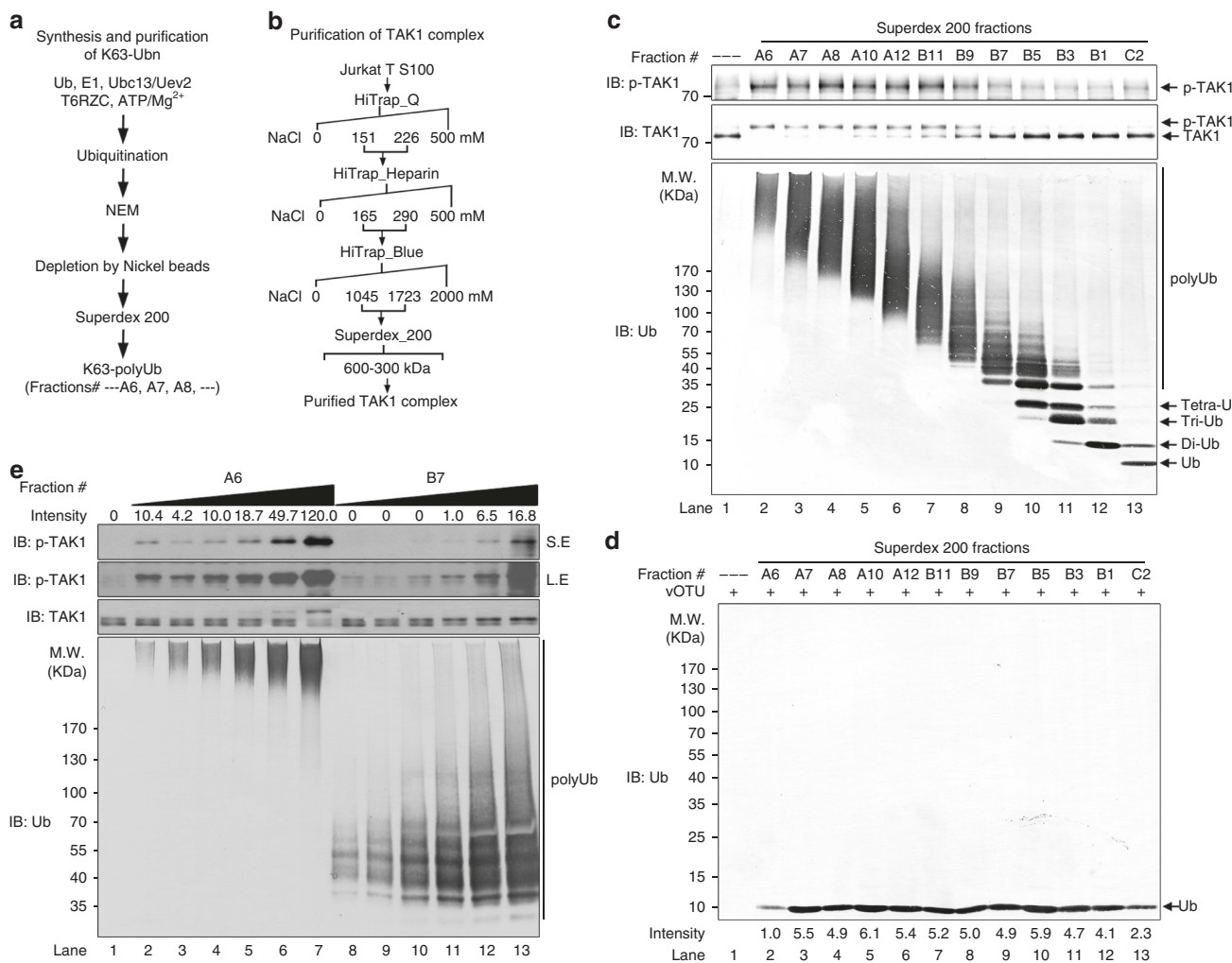

**Fig. 1** Only long free polyubiquitin chains potently activate TAK1 kinase. **a** Purification scheme of polyubiquitin (polyUb) chains. **b** Purification scheme of endogenous TAK1 kinase complex. **c** Potent TAK1 activation by high molecular size polyUb chains. Partially purified TAK1 complex as in **b** was incubated with different sizes of polyUb chains after their fractionation as in **a** in the presence of ATP for 60 min at 30 °C. Aliquots of the reaction products were analyzed by immunoblotting with the indicated antibodies. The size distribution of the polyUb chains is shown in the bottom panel. **d** The relative amounts of polyUb chains used in **c** were determined by immunoblotting with anti-Ub antibody after their complete disassembly by DUB vOTU. **e** The chain length rather than the concentration of polyUb chains contributes more to TAK1 stimulatory activity. As in **c**, TAK1 was incubated with increasing amounts of long polyUb chains (Fraction A6) or short polyUb chains (Fraction B7) to determine its dose response to polyUb chains. S.E. short exposure, L.E. long exposure, M.W. molecular weight (KDa)

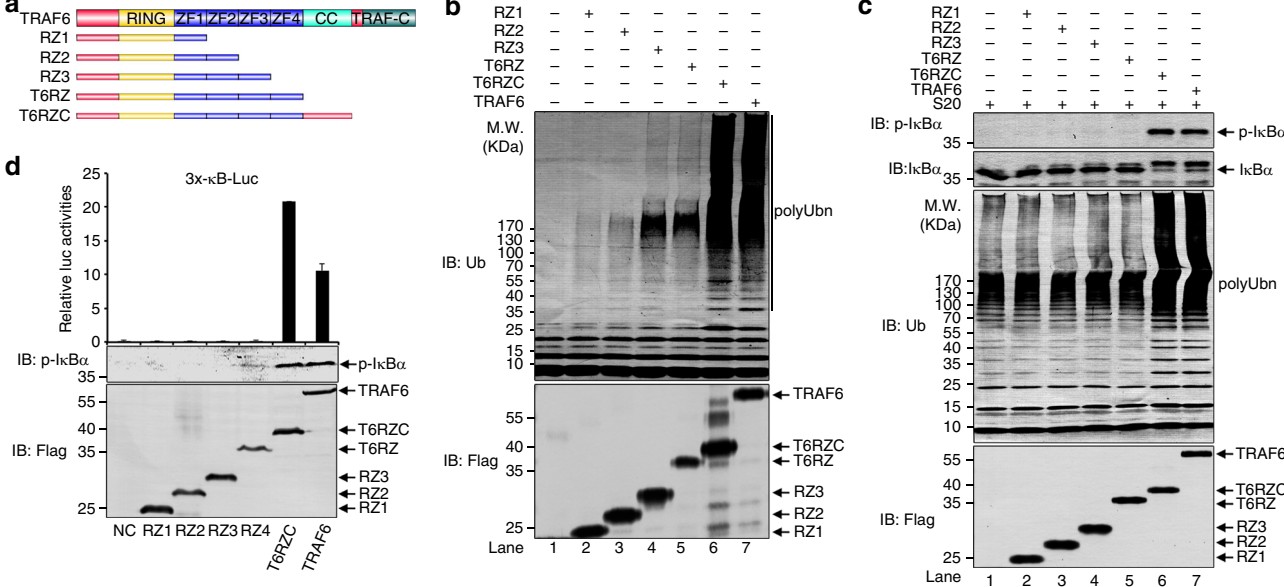

**Fig. 2** CC domain of TRAF6 is essential for long polyubiquitin chain synthesis. **a** Schematic diagram of FL TRAF6 and its truncation mutants used in this study. **b** Only FL TRAF6 and T6RZC catalyze synthesis of long polyUb chains. Flag-tagged TRAF6 or the truncation mutants were transiently expressed in HEK293T/TRAF6 KO cells and purified by using anti-Flag M2 magnetic beads (Sigma), which were then used in the ubiquitination assays. The reaction products were immunoblotted with the indicated antibodies. **c** CC domain is required for TRAF6-stimulated IKK activation in vitro. In vitro IKK activation assay was carried out with TRAF6 and its mutants purified as in **b**. The reaction products were immunoblotted with the indicated antibodies to determine activation of IKK (phosphorylation of IκBα) and synthesis of polyUb chains. **d** CC domain is required for NF-κB activation by TRAF6 in cells. TRAF6 or its mutants was transfected into HEK293 cells together with a 3 × κB luciferase reporter (3 × κB-Luc) to assess their ability to activate NF-κB. The results shown were from triplicate experiments and normalized for transfection efficiency. The data are showed as the mean ± s.d. of triplicate independent sets of experiments. The cellular extracts were also immunoblotted with the indicated antibodies

proinflammatory stimuli, the structural basis and the underlying biochemical mechanisms by which TRAF6 achieves processivity are still not well characterized. Domain analysis revealed that TRAF6 has a RING domain at its N-terminus, followed by four zinc finger (ZnF, ZnF1, 2, 3, and 4) domains, a coiled-coil (CC) domain, and a TRAF-C domain at the C-terminus[19–21]. The RING domain is responsible for direct interaction with Ubc13 and catalyzing Ub transfer for isopeptide bond formation. Crystallographic study revealed that ZnF1 also plays a structural role in supporting RING domain interaction with Ubc13[21]. Together the RING-ZnF1 fragment composes the minimal part of TRAF6 to catalyze polyUb chain synthesis with Ubc13/Uev1A in vitro. Intriguingly, even though the RING-ZnF1 fragment is sufficed to synthesize short polyUb chains, it is less efficient and fails to produce longer polyUb chains[21]. In addition, structural work revealed that Ubc13 uses the same interface for interaction with both E1 and the RING domain of TRAF6, which means Ubc13 has to dissociate from the RING domain for its recharging by Ub~E1 during the process of Ub chain extension[22], a requirement not compatible with processive Ub chain formation. These observations argue for additional structural element(s) and mechanism(s) to ensure TRAF6's processivity. The ZnF2, 3, and 4 domains are likely not the candidates since their disruption by point mutation or deletion did not cause strong defects in the IL-1, lipopolysaccharide, and RANKL signaling pathways[23], although their exact functions in TRAF6 need further analysis. Likewise, the TRAF-C domain playing a role in the processivity of TRAF6 is unlikely given that deletion of this fragment did not disrupt TRAF6 capability in processive polyUb synthesis and signaling[10, 13]. In fact, this domain serves as a binding platform mediating TRAF6 interaction with IRAK1, CD40, and many other signaling intermediates so as to receive upstream incoming activation signals thus linking TRAF6 to a particular signaling

pathway[24, 25]. In contrast, previous studies have pointed to the CC domain being essential for TRAF6 as an E3 and in signaling. For example, CC domain is important for TRAF6 auto-ubiquitination and NF-κB activation[7, 26]. Furthermore, oligo-merized form of TRAF6 correlated very well with its enhanced E3 activity and NF-κB activation[24, 27, 28], and replacement of TRAF-C domain with gyrase B dimerization domain was able to induce TRAF6 dimerization enhancing its E3 activity for TAK1-IKK activation[13]. Yet, further deletion of CC domain made TRAF6 inactive[13]. Whereas these studies illustrated the importance of CC domain for TRAF6 E3 activity and signaling, a detailed bio-chemical understanding on how CC domain contributes to TRAF6 function, especially whether and how CC domain confers processivity to TRAF6, is lacking.

Herein, we investigate systematically the structural element and the underlying biochemical mechanisms that enable TRAF6's E3 processivity when catalyzing polyUb chain synthesis with Ubc13. We find the CC domain of TRAF6 is essential for TRAF6 pro-cessivity by mediating TRAF6 oligomerization, which effectively leads to RING domain oligomerization and its subsequent interaction to multiple Ub~Ubc13 complexes and hence facil-itating processivity. More importantly, we also find that the oli-gomerized CC domain interacts with Ubc13. As a result, this interaction ensures Ubc13 association with TRAF6 while being recharged by Ub~E1 during active ubiquitination process, thus conferring processivity to TRAF6.

## Results

**Only long polyUb chains are potent TAK1 activators.** TRAF6 plays a central regulatory role in the IL-1R/TLR-induced activa-tion of TAK1–IKK–NF-κB cascade by serving as an E3 ubiquitin ligase[5, 29, 30]. Previously, we have reconstituted the TRAF6-

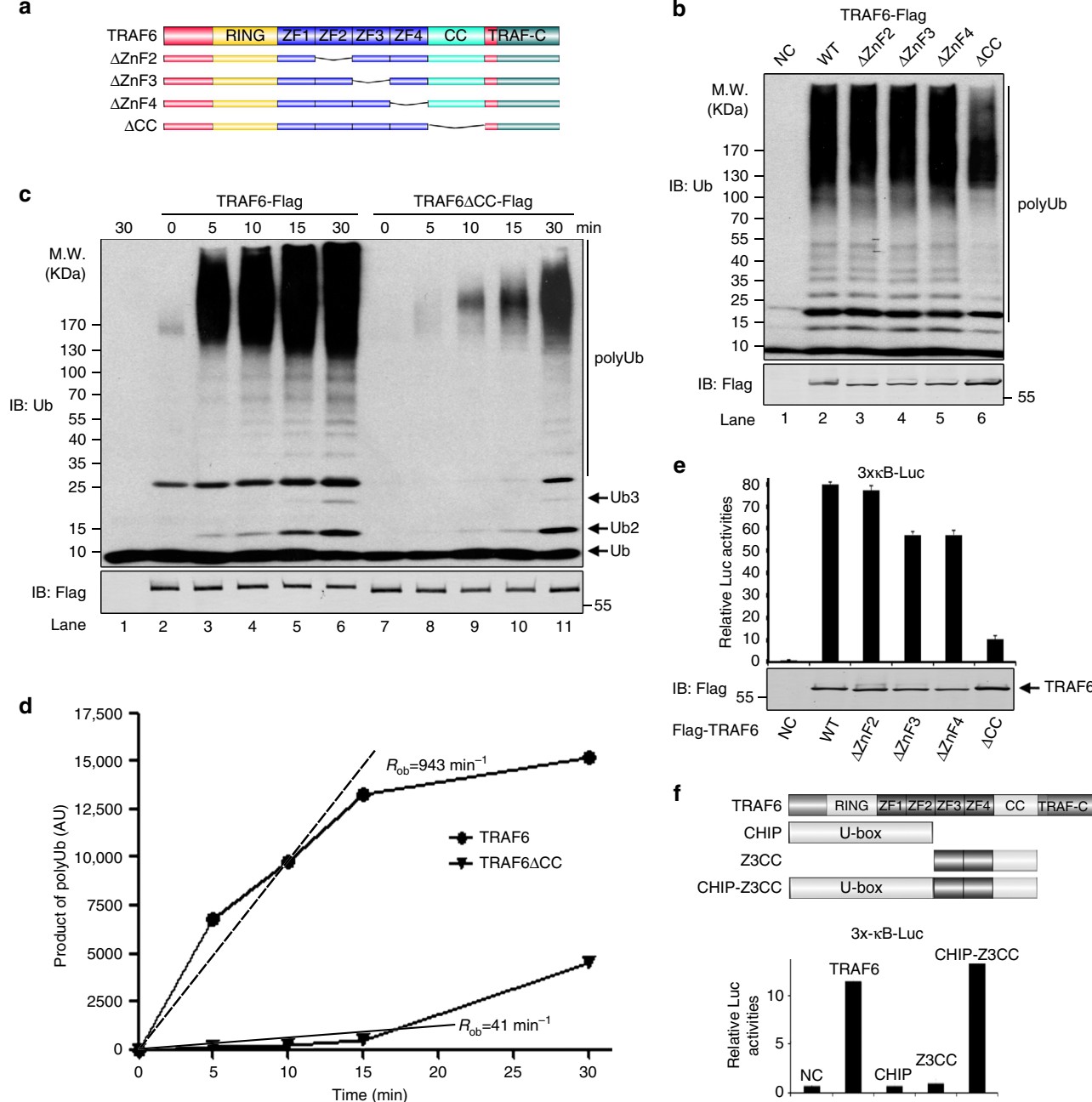

**Fig. 3** CC domain of TRAF6 is required for processive polyUb chain synthesis. **a** Schematic diagram of FL TRAF6 and its deletion mutants used in this study. **b** Deletion of CC domain (ΔCC) decreases polyUb chain assembly. Flag-tagged TRAF6 or the indicated deletion mutants were expressed in 293T/TRAF6 KO cells and purified using anti-Flag M2 magnetic beads and then used in in vitro ubiquitination assay. **c, d** Deletion of CC domain (ΔCC) decreases the rate of polyUb chain assembly. In vitro ubiquitination assay was carried out to compare time-dependent polyUb chain assembly between TRAF6 FL and ΔCC. Reaction aliquots quenched at the indicated times were evaluated by SDS-PAGE and immunoblotting (**c**), which were quantified and plotted as function of time in **d**. $R_{ob}$, arbitrary observed rate of polyUb assembly. **e** CC domain is required for NF-κB activation by TRAF6 in cells. As in Fig. 2d, but TRAF6 or its deletion mutants were used. The data are showed as the mean ± s.d. of triplicate independent sets of experiments. **f** CHIP-Z3CC activates NF-κB. Upper panel, diagram of CHIP-Z3CC fusion. Z3CC: Zinc fingers 3, 4 and CC domain of TRAF6. Bottom panel, as in Fig. 2d, reporter assays were performed with TRAF6, CHIP, Z3CC, and CHIP-Z3CC to assess their ability to activate NF-κB. The data are showed as the mean ± s.d. of triplicate independent sets of experiments

dependent TAK1 activation system using recombinant proteins, including E1, ubiquitin (Ub), Ubc13/Uev1A (E2), TRAF6 and TAK1–TAB1–TAB2 kinase complex, and found that the TRAF6-Ubc13/Uev1A system assembles unanchored K63 polyUb chains for direct TAK1 activation[10]. In the current study, by employing the same in vitro system, we further investigated the functional domains of TRAF6 and the biochemical mechanisms underlying

this polyUb chain assembly reaction. Mixing of Ub, E1, Ubc13/Uev2 (Ubc13/Uev1A and Ubc13/Uev2 both worked very well with no detectable differences with regard to polyUb chain assembly efficiency in vitro), TRAF6 (all recombinant proteins used are shown in Supplementary Fig. 1a) in the presence of ATP/Mg$^{2+}$ led to assembly of polyUb chains with different sizes ranging from di-Ub up to very high molecular weights

(Supplementary Fig. 1b, lane 2). To investigate how the size of polyUb chains influences TAK1 activation, we then scaled up the Ub chain synthesis reaction and separated the polyUb chain mixtures using size-exclusion column Superdex 200 (Sdx 200) (Fig. 1a) and tested their potency in an in vitro TAK1 activation assay (purification of endogenous TAK1 complex is shown in

Fig. 1b)[10]. Size-exclusion chromatography separates proteins according to their molecular sizes. The fractions were visualized by immunoblotting using an anti-Ub antibody (Fig. 1c, bottom panel), which revealed separation of high molecular size polyUb chains from low molecular size ones (each fraction had its own characteristic molecular size distribution). TAK1 activation was

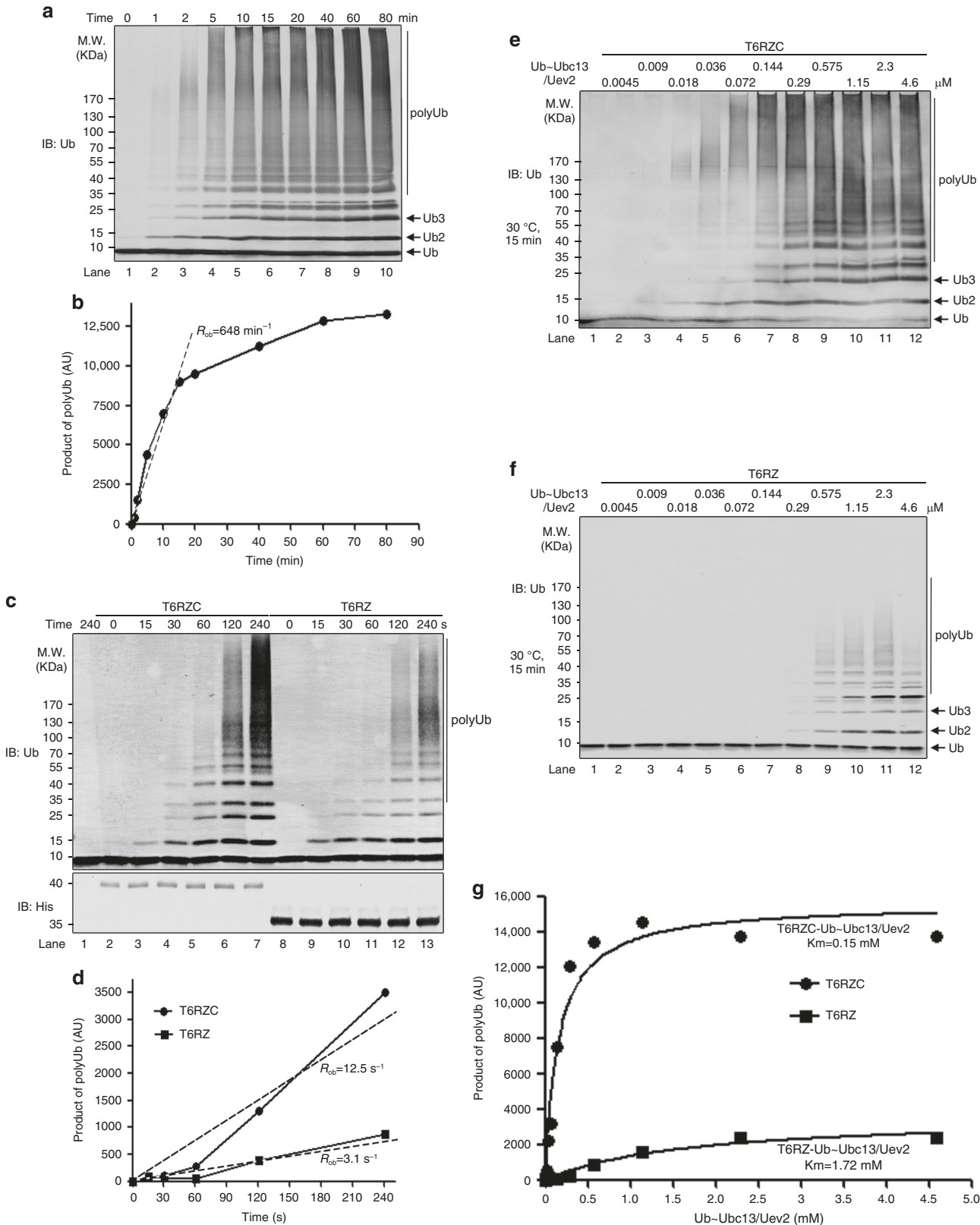

determined by its phosphorylation at Thr187[31], which revealed TAK1 activation only by polyUb chains with molecular size larger than 130 KDa (which corresponds to the first few fractions A6–B9). PolyUb chains smaller than 130 KDa (which corresponds to the remaining fractions B7–C2) displayed hardly any detectable TAK1 stimulatory activities (Fig. 1c). To determine any concentration differences of polyUb chains, we disassembled same volumes of them from these fractions with the viral OTU (vOTU) into mono-Ub and visualized the resulting products by an anti-Ub immunoblotting, which revealed similar amounts in terms of mono-Ub across these fractions except that it is lower in A6 (Fig. 1d). vOTU is an OTU-containing DUB encoded by Crimean-Congo hemorrhagic fever virus and can disassemble all the seven Ub linkages[10, 32]. These observations suggest that the longer the polyUb chains, the more potent their TAK1 stimulatory activities. In other words, it is the chain length rather than the concentration of polyUb chains that contributes more to their kinase stimulatory activities.

To further confirm our claim, we did a dose-response analysis on TAK1 activation by comparing fractions A6 and B7, which illustrated that polyUb chains from A6 displayed appreciable activities in a dose-dependent manner even at a very low concentration, whereas B7 exhibited very weak activities (Fig. 1e). Furthermore, the weak TAK1 activation at high concentration from B7 fraction might be contributed from residual amount of the long polyUb chains (Fig. 1e). The polyUb chain size distribution in fraction A6 is above 250 KDa, whereas that is between 35 and 100 KDa in B7. In terms of mono-Ub concentration, A6 is about 5-fold lower than B7 (Fig. 1d, lanes 2 vs. 9), while its TAK1 kinase stimulatory activity is about eight times more potent than B7 (Fig. 1e, lanes 2–7 vs. 8–13). Altogether, these findings support our claim that longer polyUb chains are more potent TAK1 kinase activators. The requirement of very long polyUb chains for potent TAK1 activation implies a need for TRAF6 to be a processive Ub ligase.

**Long polyUb chain assembly requires CC domain of TRAF6**. To investigate the structural basis for efficient assembly of long polyUb chains by the TRAF6-Ubc13/Uev2 system, we focused on TRAF6 to see which domain(s) are required for this activity. We systematically generated a set of truncation mutants from its C-terminus (Fig. 2a), and purified mutant proteins after transient overexpression in 293T cells and then tested their activities using our in vitro polyUb reaction assay. Full-length (FL) TRAF6 and the T6RZC (in which only the C-terminal TRAF-C domain was removed) catalyzed comparable synthesis of long polyUb chains (Fig. 2b). Further truncation toward the N-terminus resulted in shortened proteins that catalyzed assembly of polyUb chains with lower molecular weights. These data illustrate that the CC domain of TRAF6 is required for efficient long polyUb chain assembly. Consistently, in the in vitro S100-based IKK activation assay[5], only FL TRAF6 and T6RZC activated the TAK1–IKK–IκBα, which can be detected by a phospho-specific antibody for phosphorylated IκBα and its slow mobility shift on SDS-PAGE (Fig. 2c). Immunoblotting of the reaction products

also revealed only the FL TRAF6 and T6RZC produced longer polyUb chains, which correlated well with TAK1–IKK–IκBα activation. Similarly, it was also only the FL TRAF6 and T6RZC, but not the other truncation mutants, that stimulated the $3 \times \kappa$B-Luc reporter activities after transient expression in 293T cells (Fig. 2d).

To further evaluate the importance of CC domain in the assembly of long polyUb chains, we systematically generated a set of internal deletion mutants (Fig. 3a) and tested their activities in our polyUb chain assembly assay. Deletion of ZnF2, ZnF3, or ZnF4 did not lead to appreciable damage to their activities since they exhibited comparable activities to that of FL TRAF6 in promoting polyUb chain assembly (Fig. 3b). In contrast, deletion of CC domain (ΔCC) decreased polyUb chain assembly, especially those with very high molecular weight (Fig. 3b). Kinetic comparison between TRAF6 and ΔCC revealed that ΔCC synthesized polyUb chains in a slower rate than FL TRAF6 did (Fig. 3c). The protein quantification in Fig. 3c revealed the relative rate of polyUb chain assembly differs by ~20-fold between FL TRAF6 and ΔCC in the first 15 min ($R_{ob} = 943$ min$^{-1}$ vs. $R_{ob} = 41$ min$^{-1}$, Fig. 3d). Consistently, only ΔCC mutant displayed activity reduction in the $3 \times \kappa$B-Luc reporter assay (Fig. 3e), as was also reported by Wang et al.[7]. Interestingly, grafting the ZnF3-ZnF4-CC (Z3CC) fragment to the U-box-containing E3 CHIP (Fig. 3f, top panel), which by itself is not able to activate NF-κB, rendered CHIP to be a potent activator of $3 \times \kappa$B-Luc reporter (Fig. 3f; Supplementary Fig. 2a), implying the CC domain is sufficient to redirect its processivity-promoting activity to another E3 ligase. Of note, in the in vitro S100-based IKK assay, ΔZnF3, ΔZnF4, and ΔCC were all defective in activating IKK (Supplementary Fig. 2b). However, the activities of both ΔZnF3 and ΔZnF4 but not the ΔCC were readily reversed by the addition of ubiquitin aldehyde (Ubal), a Ub derivative with broad spectrum of inhibitory activities on DUBs[33] (Supplementary Fig. 2b). One possible explanation is that the biochemical function of ZnF3 and ZnF4 might be to counteract DUB activities therefore facilitating assembly of long polyUb chains. Consistently, time-course analysis of polyUb chain assembly reactions with purified proteins in which there was no detectable contamination of DUBs revealed that ΔZnF3 and ΔZnF4 were able to synthesize polyUb chains comparable to FL TRAF6 did, although with some reduced rate in the beginning (Supplementary Fig. 2c).

Together, these systematic studies on truncation and deletion mutants demonstrate the importance of the CC domain in conferring processivity to TRAF6 E3 activity, without which the intrinsic E3 processive activity of TRAF6 is compromised.

**CC domain enables processive polyUb chain synthesis**. To investigate to what extent CC domain contributes to processive polyUb synthesis, we analyzed the kinetics of polyUb chain assembly reaction by the TRAF6-Ubc13/Uev2 system. We observed that polyUb synthesis by TRAF6-Ubc13/Uev2 is a rapid event in that Ub chains, especially those long polyUb chains, after an initial short time delay, started to form and accumulated

**Fig. 4** CC domain enables processive polyUb chain synthesis by TRAF6. **a**, **b** Processive polyUb chain assembly by TRAF6. Time-course analysis of polyUb chain synthesis by T6RZC was performed as indicated and shown in **a**. The corresponding quantification was plotted as function of time in **b**. $R_{ob}$, arbitrary observed rate of polyUb assembly. **c**, **d** Lack of CC impairs polyUb chain synthesis by TRAF6. Time-course analysis of polyUb chain synthesis was performed to compare T6RZC and T6RZ. Time-dependent accumulation of polyUb chains was shown in **c** and quantified in **d**. It is needed to note we used a lot more of T6RZ in order to enhance the rate by T6RZ so we could visualize the products and do the reliable quantification. $R_{ob}$, arbitrary observed rate of polyUb assembly. **e**–**g** Km determination of Ub~Ubc13 for T6RZC and T6RZ. Pre-charged Ub~Ubc13/Uev2 were titrated into T6RZC or T6RZ and incubated at 30 °C for 30 min. The products were immunoblotted (**e**, **f**), quantified and plotted as function of Ub~Ubc13/Uev2 in **g**. The Km values were estimated by fitting two independent data sets to Michaelis–Menten kinetics

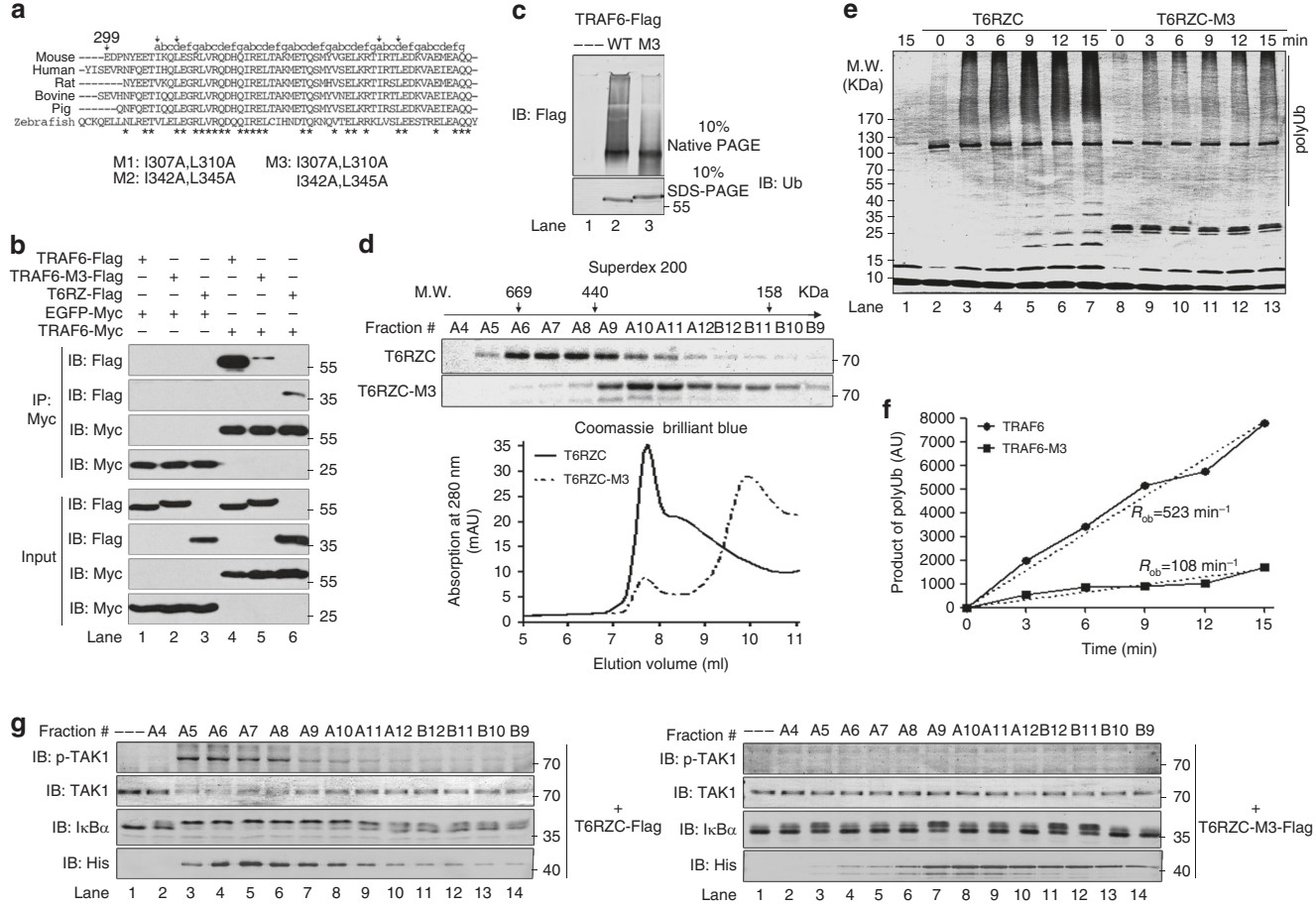

**Fig. 5** CC domain mediates TRAF6 intermolecular interaction and oligomerization. **a** Amino acid sequence alignment of TRAF6 CC domains from different species. The heptad repeats are denoted with "a-b-c-d-e-f-g". Arrows indicate the mutation residues used in the study. **b** CC domain mediates TRAF6 self-association. Flag-tagged TRAF6, mutant M3, or T6RZ were co-transfected with Myc-tagged TRAF6 or with Myc-tagged EGFP into HEK293T cells and subjected to immunoprecipitation with anti-Myc antibodies. The immunoprecipitated complexes (upper panels) and cell lysates (down panels) were immunoblotted with the indicated antibodies. **c** CC domain mediates TRAF6 oligomerization. TRAF6-Flag and TRAF6-M3-Flag were expressed in HEK 293T cells and purified using anti-Flag M2 magnetic beads. The eluted proteins were analyzed by 10% native gel. **d** CC domain mediates TRAF6 oligomerization. T6RZC and T6RZC-M3 were purified from Sf9 cells and separated by size-exclusion column Superdex 200. Aliquots from each fraction were separated by 10% SDS-PAGE and detected by Coomassie blue staining. Superimposed gel filtration profiles of T6RZC and T6RZC-M3 are shown below. **e, f** Disruption of CC oligomerization impairs polyUb chain synthesis. Time-course analysis of polyUb chain synthesis was performed to compare T6RZC and T6RZC-M3. Time-dependent accumulation of polyUb chains was shown in **e** and quantified in **f**. $R_{ob}$, arbitrary observed rate of polyUb assembly. **g** Only the high molecular size TRAF6 can activate TAK1 and IKK. Aliquots from **d** were subjected to in vitro IKK activation assay. Activation of TAK1 and IKK was determined with the indicated antibodies

almost linearly in the first 15 min. After which it slowed down. Nevertheless, the total polyUb chains still kept accumulating (Fig. 4a, b; Supplementary Fig. 3a, b). (It is worth to point out that throughout this study, we noticed our in vitro polyUb synthesis reaction rates always followed a linear way during the first 15 min and the rates then started to decrease. So our calculations were all based on data collected within the first 15 min.) We reason that the initial delay, after mixing of all the components, could be due to the Ub activation by E1 and its subsequent transfer to Ubc13 to form Ub~Ubc13 complex. This is also consistent with our two-step reaction mode, in which we first mixed Ub, E1, Ubc13/Uev2, and ATP together for 5 min and then T6RZC was added (Supplementary Fig. 3c). In this way, efficient polyUb chains were readily accumulated in as short as 5 s, the shortest time period we could handle with reliability (Supplementary Fig. 3d).

Using this method, we monitored the polyUb chain synthesis reaction kinetics of T6RZC and T6RZ. Side-by-side comparison demonstrated that T6RZC catalyzed polyUb chain accumulation faster than T6RZ did (Fig. 4c), as it could be seen from the

estimated rates. As such, the difference between T6RZC and T6RZ in terms of the observed rates was about 4-fold ($R_{ob} = 12.5\ s^{-1}$ vs. $R_{ob} = 3.1\ s^{-1}$, Fig. 4d). Consequently, we conclude that the lack of CC domain reduced the rate of polyUb chain extension.

To measure the Km values of T6RZC or T6RZ with regard to their catalysis of polyUb synthesis, we titrated Ub~Ubc13/Uev2 with T6RZC or T6RZ, respectively (Fig. 4e, f). Thus, we estimated the Km for Ub~Ubc13/Uev2-T6RZC is about 0.15 μM and the Km for Ub~Ubc13/Uev2-T6RZ is about 1.72 μM (Fig. 4g), which is a difference about 11.5-fold. This is also consistent with the CC domain being important in TRAF6 catalysis of polyUb chain formation and illustrates a decrease in catalysis by TRAF6 and Ub~Ubc13 without the CC domain. Previous studies have determined the steady-state Km for Ub against Ubc13/Mms2 is about 165 μM and the cellular free mono-Ub concentration is the range of 10–20 μM[34, 35], which means less than 10% of total cellular Ubc13 would be in the activated form of Ub~Ubc13 under steady-state. By using quantitative immunoblotting, we

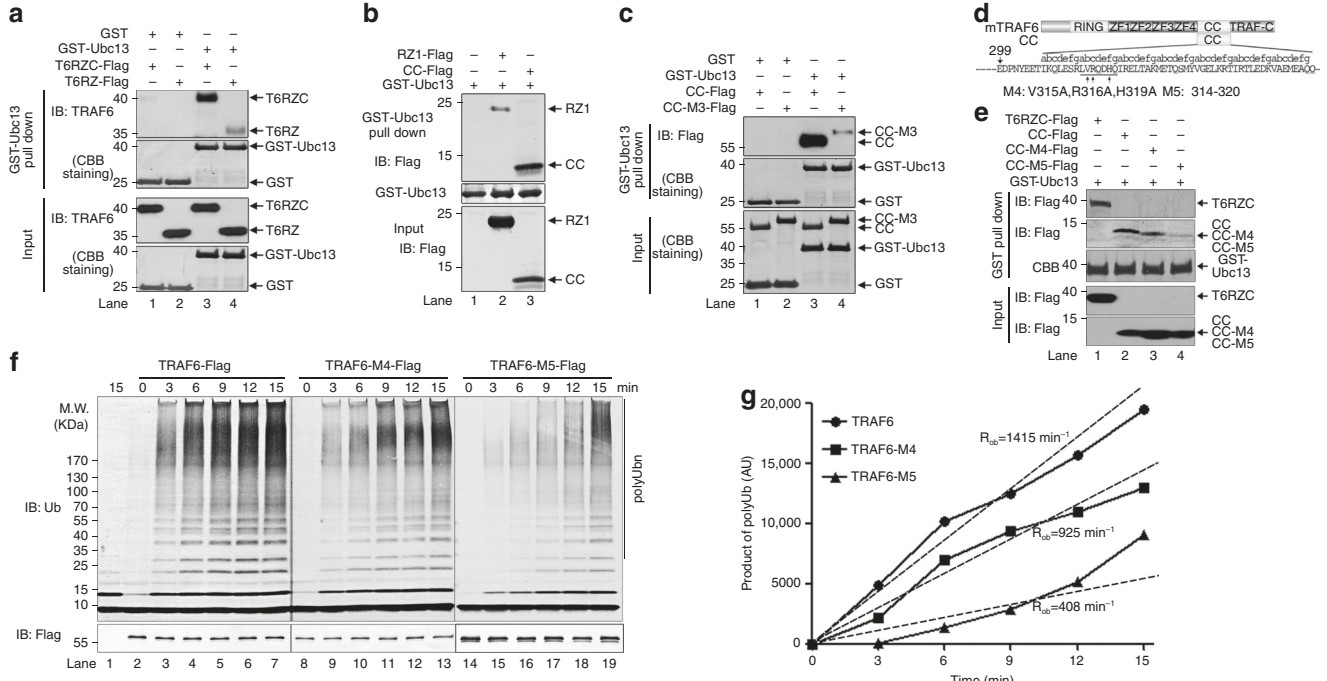

**Fig. 6** Oligomerized CC domains interact with Ubc13 contributing to processivity of TRAF6 E3 activity. **a** CC domain contributes to the interaction between TRAF6 and Ubc13. GST or GST-Ubc13 was incubated with purified T6RZC or T6RZ proteins and subjected to GST pull-down assays. Bound proteins were analyzed by immunoblotting with the indicated antibodies. **b** CC domain binds more Ubc13 than RZ1 does. HEK293T/TRAF6 KO cell lysates expressing Flag-tagged RZ1 or CC were mixed with GST-Ubc13 and subjected to GST pull-down assay. Bound products were immunoblotted with the indicated antibodies. **c** Oligomerized CC domains bind more Ubc13. GST or GST-Ubc13 was mixed with purified CC-Flag or CC-M3-Flag and subjected to GST pull-down assay. Bound proteins were immunoblotted with the indicated antibodies. **d** TRAF6 CC domain mutants M4 and M5 are depicted. **e** Impaired interaction between CC mutants and Ubc13. HEK293T/TRAF6 KO cell lysates expressing T6RZC-Flag, CC-Flag, CC-M4-Flag, or CC-M5-Flag were mixed with GST-Ubc13 and subjected to GST pull-down assay. Bound products were immunoblotted with the indicated antibodies. **f, g** TRAF6 mutants M4 and M5 display decreased rate of polyubiquitin chain assembly. Time-course analysis of polyUb chain synthesis was performed to compare TRAF6, TRAF6-M4, and TRAF6-M5. Time-dependent accumulation of polyUb chains was shown in **f** and quantified in **g**. $R_{ob}$, arbitrary observed rate of polyUb assembly

determined the cellular Ubc13 concentration in 293T cells is ~4.9 μM, which is consistent with what was obtained by quantitative mass spectrometry[36]. Consequently, the concentration of active Ub~Ubc13 complex is about 0.5 μM under steady-state condition in 293T cells, which means 80% of TRAF6 would be engaged with Ub~Ubc13 complex for active polyUb chain synthesis, once it is activated. But without CC domain, the TRAF6 fraction to be engaged would drop to less than 20%.

**CC domain mediates TRAF6 oligomerization.** We next were to address how the CC domain could confer processivity to TRAF6 E3 activity. Previous reports have shown that only high molecular weight TRAF6 oligomers are active in S100-based IKK activation assay[27]. Therefore, we tested if CC domain plays a major role in mediating TRAF6 oligomerization. Multiple programs that we used[37, 38] predicted that the TRAF6 CC domain is a parallel left-handed CC and there are at least seven bona fide repeats of the heptad unit (depicted as a-b-c-d-e-f-g) (Fig. 5a). Residues at "a" and "d" positions are in general nonpolar mediating CC domain association through hydrophobic and van der Waals interaction[39, 40]. We thus mutated the highly conserved residues at "a" and "d" positions in the CC domain to Ala residues[41] and constructed three mutants M1 (I307A, L310A), M2 (I342A, L345A), and M3, in which M3 combines the mutation sites of M1 and M2 (Fig. 5a). When compared to FL TRAF6, M1 and M2 mutants displayed mildly, while M3 more reduced activities in the 3 × κB-Luc reporter assay (Supplementary Fig. 4a), and therefore M3 was chosen for further analysis. As expected, Flag-tagged M3 mutant protein, similar to T6RZ, immunoprecipitated less Myc-

tagged FL TRAF6 (Supplementary Fig. 4b) and vice versa (Fig. 5b). Importantly, FL TRAF6 but not M3 mutant protein formed very high molecular weight species as it can be seen from the native PAGE gel, which implies that TRAF6 can oligomerize via the CC domain giving rise to high molecular weight oligomers (Fig. 5c). Similarly, we found by size-exclusion chromatography that FL TRAF6 eluted in the void volume, whereas M3 mutant protein eluted in the later fractions with the peak fraction corresponding to 300 KDa (Fig. 5d). Of note, other parts of TRAF6 (for example, its RING-RZ1 fragment[21]) should also play roles in mediating their self-interaction since the size of M3 is far above its monomer size of ~60 KDa. Nevertheless, the M3 mutant protein displayed decreased activity in the polyUb synthesis reaction with apparent rate difference from FL TRAF6 by about 5-fold (Fig. 5e, f; Supplementary Fig. 4c). Consequently, M3 displayed reduced activities in the in vitro TAK1 (Supplementary Fig. 4d) and IKK activation assays (Fig. 5g; Supplementary Fig. 4e), respectively, in which it did not catalyze efficient synthesis of polyUb chains (Supplementary Fig. 4d, e). Taken together, these data demonstrate a chain of consequential events involving the CC domain, its oligomerization leading to high molecular weight TRAF6 oligomers, to the processive assembly of high molecular weight polyUb chains, and to kinase activation.

**Oligomerized CC domain interacts with Ub~Ubc13.** We next setup to investigate why oligomerized TRAF6 can synthesize polyUb chains processively. We noticed when we performed GST-based pull-down experiments that GST-Ubc13 (Supplementary Fig. 5a) pulled down more T6RZC than T6RZ (Fig. 6a).

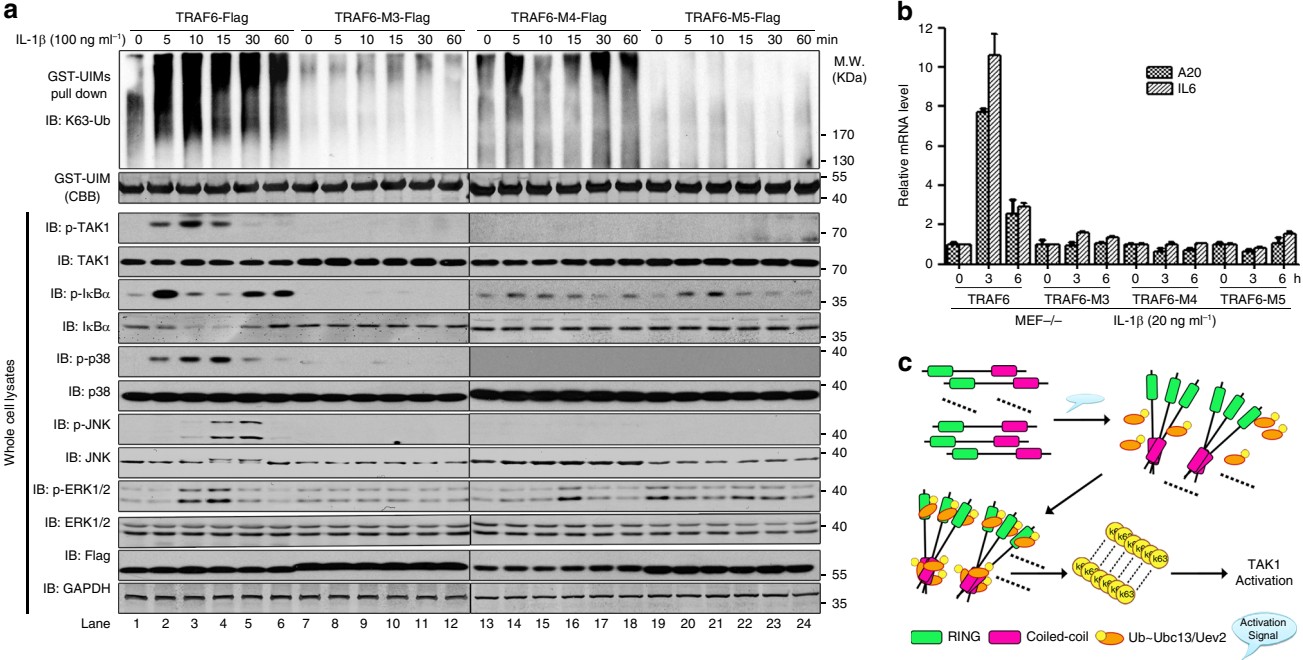

**Fig. 7** CC domain-mediated oligomerization and interaction with Ubc13 are critical for the biological functions of TRAF6. **a** TRAF6 mutants M3, M4, and M5 fail to rescue the activation of TAK1, IKK, and MAPK kinases in the IL-1R signaling pathway. TRAF6 KO MEF cells reconstituted with TRAF6 WT or M3, M4, M5 mutants were stimulated with IL-1β for the indicated duration of time. Cell lysates were analyzed by immunoblotting with the indicated antibodies. To capture K63-linked polyubiquitin chains, cell lysates were subjected to GST-PAP80-UIMs-based pull down. Bound polyUb chains were separated on 6–18% SDS-PAGE and immunoblotted with an anti-K63 Ub antibody. **b** TRAF6 mutants M3, M4, and M5 fail to rescue the activation of NF-κB. As in **a**, but cells were untreated or treated with IL-1β for 3 and 6 h, respectively. Total RNAs were extracted, and qRT-PCR was performed to measure the induction of IL-6 and A20. The data are showed as the mean ± s.d. of triplicate independent sets of experiments. **c** Model depicting the contribution of CC domains to TRAF6 processivity. Upon receiving upstream activation signals, CC domains start to trimerize, which, on the one hand, leads to TRAF6 trimerization and effective RING domain trimerization and their binding to Ub~Ubc13/Uev2; on the other hand, trimerized CC domains themselves also bind to Ub~Ubc13/Uev2. These two events together contribute to processive polyUb chain assembly

One potential explanation would be CC domain, after oligomerization, interacts with Ubc13 directly, which we examined. We found that GST-Ubc13 indeed pulled down the CC domain (Fig. 6b, lane 3). As expected, GST-Ubc13 also pulled down RZ1 but less than CC domain, implying Ubc13 interaction with CC with stronger affinity than with RZ1 (Fig. 6b). CC domain also co-immunoprecipitated endogenous Ubc13 (Supplementary Fig. 5b) and shifted Ubc13 from low molecular size fractions to high molecular size fractions on the size-exclusion column Superose 6 (Supplementary Fig. 5c). Since T6RZC displayed processivity whereas T6RZ did not (Fig. 4e, f), we tested whether oligomerization would affect CC–Ubc13 interaction and found this is the case since GST-Ubc13 pulled down much more CC than CC-M3 (Fig. 6c).

To better understand the interaction between the oligomerized CC domain and Ubc13 and its effect on TRAF6 processivity, we generated two mutants on the CC domain, M4 (V315A, R316A, H319A mutation, which are all highly conserved in TRAF6 homologs and at positions "b", "c", and "f", respectively) and M5 (deletion of one complete heptad unit, amino acids 313–319) (Figs. 5a, 6d). Mutated residues in M4 lie on the helical surface that is exposed to the solvent[40, 42]. We surmise that this surface mediates the CC domain interaction with other molecules. Pertaining to the discussion here, the surface would mediate CC domain interaction with Ubc13. As expected, both M4 and M5 mutants did not affect TRAF6 inter-molecular interaction in that Flag-tagged M4 and M5, as wild-type (WT) did, immunoprecipitated comparable amount of Myc-tagged TRAF6 WT (Supplementary Fig. 5d), formed high molecular weight species on native gel (Supplementary Fig. 5e) and on size-exclusion column

Superdex 200 (Supplementary Fig. 5f). However, M4 and M5 mutants showed reduced interaction with Ubc13 given that GST-Ubc13 pulled down less CC-M4 and even much less CC-M5 than WT (Fig. 6e). Accordingly, in the ubiquitination assay, both M4 and M5 mutants exhibited reduced rate of polyUb assembly (by about 1.5 and 3 folds, respectively), with M5 performing the worst (Fig. 6f, g). Consequently, M4 and M5 displayed reduced activities not only in the 3 × κB-Luc reporter assay (Supplementary Fig. 5g) but also in the S100-based IKK activation assay (Supplementary Fig. 5f, upper panel), which again correlates well with their respective defects in polyUb chain synthesis (Supplementary Fig. 5h, bottom panel).

During ubiquitination reaction, Ubc13 is loaded with Ub forming Ub~Ubc13 intermediate complex, and this is the major entity to interact with TRAF6. We therefore examined TRAF6–Ubc13 interaction after Ubc13 was loaded with Ub to understand how activated Ub~Ubc13 plays a role in TRAF6–Ubc13 interaction. Purposely, we generated a Ubc13 C87S mutant, which is known to form Ub~Ubc13 more stably than WT do[43], and thus more amenable for biochemical assays such as GST pull down. Thus, we found that both Ubc13~Ub and Ubc13 interacted with T6RZC; however, they showed less affinity toward T6RZC (I72D, L74H), a mutant with two residue mutations in the RING domain known to be involved in direct interaction with Ubc13[21]. Removal of CC domain (T6RZ) reduced interaction toward Ubc13 or Ub~Ubc13 indiscriminately, which is regardless of the ubiquitin loading on Ubc13. Combination of CC deletion and RING domain (I72D, L74H) mutations completely abolished the interaction between TRAF6 and Ubc13~Ub or Ubc13 (Supplementary Fig. 5i). These data

further illustrate that both CC and RING domains of TRAF6 can independently interact with Ubc13~Ub/Ubc13 and CC domain contributes more to their interaction.

Taken all the data together, they demonstrate that oligomerized CC domain, through its interaction with Ubc13, enables processive polyUb chain assembly by TRAF6.

**CC–Ubc13 interaction critical for TRAF6 biological functions**. Finally, we tested our in vitro findings in cell-based assays by reconstituting TRAF6 knockout (KO) mouse embryonic fibroblast (MEF) cells with FL TRAF6 WT, and the mutants M3, M4, M5, respectively. We initially confirmed that TRAF6 WT is able to restore IL-1β-induced NF-κB signaling events to the level comparable to MEF WT cells (Supplementary Fig. 6a, b). Thus, upon IL-1β stimulation, TRAF6 WT restored phosphorylation and activation of TAK1 and the downstream signaling events (IκBα phosphorylation and degradation, activation of the three MAPK pathways) (Fig. 7a, bottom panels). In contrast, all the three mutants we tested failed to rescue these signaling events (Fig. 7a, bottom panels). Consistently, only TRAF6 WT but not the three mutants restored induction of NF-κB target genes IL-6 and A20 (Fig. 7b). In addition, we reconstituted the TRAF6 KO MEF cells with T6RZC-gyrB WT and M3 mutant in which their respective TRAF-C domains were replaced with the dimerization domain of a bacterial DNA gyrase B subunit[13, 44, 45]. The dimerization domain of gyrase B can be induced to form dimers by the small compound Coumermycin A1[13, 44, 45]. In this way, TRAF6 can be activated to initiate signaling by Coumermycin A1[13, 44, 45]. Upon treatment with Coumermycin A1, T6RZC-gyrB but not the mutant M3 rescued the signaling events and induction of IL-6 and A20 downstream of TRAF6 (Supplementary Fig. 6c, d), further suggesting TRAF6 oligomerization by CC domain plays crucial roles for its proper functions and TRAF-C domain is indeed not essential for its E3 ligase activities.

Remarkably, during the immunoprecipitation procedure using GST-UIM (a high specificity and affinity binding motif of K63 polyUb chains[46]), we were able to pull down significant amount of polyUb chains upon IL-1β stimulation from cells reconstituted with TRAF6 WT (Fig. 7a, top panel). The production of the polyUb chains was correlated very well with TAK1 activation and signaling. However, in contrast to TRAF6 WT, we did not pull down decent amount of polyUb chains in M3, M4, or M5 reconstituted cells after IL-1β treatment (Fig. 7a, top panel). These data support a crucial role of CC domain-mediated TRAF6 oligomerization and its interaction with Ubc13 in TRAF6's biological functions in cells.

## Discussion
Our data have clearly demonstrated that only the high molecular weight polyUb chains can activate TAK1 very potently, whereas the low molecular weight ones possess hardly any activity even at a very high concentration. Exactly why only long polyUb chains are active for TAK1 stimulation is not clear at present. Previously, we provided experimental evidence to show that TAB2 subunit of the TAK1 kinase complex is the polyUb chain acceptor that leads to TAK1 dimerization/oligomerization and subsequent TAK1 autophosphorylation and activation[10]. We reason that the longer the polyUb chains, the higher their affinity with TAB2 than shorter polyUb chains. Thus, long polyUb chains could bind more TAK1 complex molecules and by bringing them together can effectively increase TAK1 local concentration, which then translates into a potent TAK1 activation. We also reason that long polyUb chains might adopt higher-order structures after reaching a certain length. Consistent with this assumption, previous studies demonstrated that mono-ubiquitin molecules can interact

with each other non-covalently[47]. Therefore, it is conceivable that Ub moieties of K63 polyUb chains can also interact with one another non-covalently, which might mediate formation and stabilization of higher-order structures. With new powerful imaging tools such as atomic force microscopy and Cryo-EM becoming more readily accessible, future efforts will be directed on investigation of higher-order structures of long polyUb chains.

Facing the challenge to synthesize very long polyUb chains in a timely manner when cells are challenged with the need to activate TAK1 promptly, the TRAF6-Ubc13 ubiquitination system has to assemble K63 polyUb chains in a processive way. Our findings about the CC domain revealed that TRAF6 explores two mechanisms to achieve processivity. On the one hand, the CC domain promotes TRAF6 oligomerization. In this way, all the RING domains in a single oligomer are grouped to function like a single entity binding numerous Ub~Ubc13, which can take turns as Ub donors. Functionally, the simultaneous binding of multiple Ub~Ubc13 may be considered as a "pseudo" single Ub~Ubc13 that is undergoing active "discharging-recharging" cycles without dissociation from the RING domain. Therefore, it is conceivable that the larger the TRAF6 oligomer, the more RING domains are present and more Ub~Ubc13 can be recruited to these RING domains, which could lead to longer polyUb chain assembly by a single binding event. On the other hand, after oligomerization CC domain itself binds Ubc13 (or Ub~Ubc13). By this way, the spent Ubc13 can be re-loaded with Ub without complete dissociation from TRAF6, that is, through the "on-site" recharging, which can effectively promote the rate of polyUb chain synthesis. Usage of this mechanism has been observed in the cases of Ube2g2/gp78 and Ubc2/Rad6/Ubr1[48, 49]. TRAF6, simply through CC domain-mediated oligomerization, combines these two mechanisms together, thereby maximizing its processive assembly of polyUb chains. The importance of these two mechanisms to enable TRAF6's processivity is well demonstrated by our analyses of mutants M3 and M4/M5. The M3 mutant impairs CC domain self-oligomerization, thus preventing TRAF6 from higher-order oligomerization. This in turn leads to the impairment of TRAF6's processivity, which well speaks for the importance of the first mechanism. The M4 and M5 mutants, which contain mutation sites on the surface of the CC domain without defects in oligomerization, showed impaired interaction with Ubc13/Ubc13~Ub and illustrate the critical role of the second mechanism in conferring processivity to TRAF6 E3 ligase activity. Since M3 and M4/M5 are all defective in efficient polyUb chain assembly, it also suggests that for TRAF6, both mechanisms have to be coupled/combined to achieve its proper processivity.

The interaction of CC domain with Ubc13 and its importance for TRAF6 processivity is interesting. We provided three lines of evidence to illustrate their interaction. (1) GST-Ubc13 is able to pull down CC domain; (2) CC domain can immunoprecipitate Ubc13 in cells; (3) CC domain and Ubc13 co-migrate on size-exclusion column Superose 6. Our data also demonstrated the importance of CC oligomerization in priming its interaction with Ubc13, for oligomerization-defective mutant CC-M3 displays weaker interaction with Ubc13 than CC WT did. This observation also implies that, after oligomerization, a new composite binding surface contributed from each subunit might be formed to mediate CC domain interaction with Ubc13/Ub~Ubc13. Consistent with this assumption, two mutants in this study, CC-M4 and CC-M5 made with an intention of mutating surface residues, are intact in self-oligomerization but display defect in Ubc13 interaction. These data also suggest this region of the CC domain could be the direct binding site for Ubc13. Co-crystallization of CC–Ubc13 complex will provide details about their direct binding sites and mode of interaction.

In summary, by using polyubiquitination assays, we characterized the processivity of E3 ligase TRAF6 and linked its processivity to the CC domain (Fig. 7c). As such, CC domain not only promotes TRAF6 oligomerization, but also the oligomerized CC domain binds Ubc13 effectively. Together, these two mechanisms confer processivity to TRAF6 E3 ubiquitin ligase activity.

## Methods

**Cell lines and reagents**. TRAF6 KO and matched WT MEFs were kindly provided by Dr H. Xiao (Institute Pasteur of Shanghai, China). HEK293T, HeLa, and Jurkat T cells were originally from ATCC. Mouse antibody against ubiquitin (sc-8017, dilution 1:1000) was purchased from Santa Cruz Biotechnology; Rabbit antibodies against p-TAK1 Thr187 (#4536, dilution 1:1000), p-p38 Thr180/Tyr182 (#9211, dilution 1:1000), p-JNK Thr183/Tyr185 (#4668, dilution 1:1000), p38 (#8690, dilution 1:1000), JNK (#9252, dilution 1:1000), p-ERK1/2 (#4370, dilution 1:1000), and K63-linkage-specific polyubiquitin chains (#5621, dilution 1:1000) were purchased from Cell Signaling; Mouse monoclonal antibody against p-IκBα Ser32/36 (#9246, dilution 1:1000) was purchased from Cell Signaling. Rabbit monoclonal antibodies against TRAF6 (ab33915, dilution 1:2000), TAK1 (ab109526, dilution 1:10,000), and Ubc13 (ab109286, dilution 1:5000) were obtained from Abcam; Mouse antibodies against Flag-tag (M20008, dilution 1:5000), GST-tag (M20007, dilution 1:5000), His-tag (M20001, dilution 1:5000), GAPDH (M2006, dilution 1:5000) were purchased from Abmart. Goat anti-rabbit and goat anti-mouse IgG-conjugated horseradish peroxidase were purchased from Promega. M2-conjugated magnetic beads, N-ethylmaleimide (NEM), and other chemicals were purchased from Sigma. Nickel-agarose beads (#88221) and Immobilized Glutathione (#15160) were purchased from Thermo Scientific. Amylose resin (#8021L) was purchased from New England Biolabs. Ubal (Ub-H) was purchased from Boston Biochem. Recombinant mouse IL-1β was purchased from Sino Biological.

**Plasmids and recombinant proteins**. Mammalian expression plasmids encoding mouse TRAF6 and its variants were PCR amplified and ligated into pXC-Flag or pXC-MBP-Flag retroviral vectors modified from pMX vector. For protein expression in Sf9 cells, cDNAs encoding human E1 and TRAF6 were subcloned into pFastBac-HTB vector. cDNAs encoding Ubc13(UBE2N), Ubc13(C87S), Uev2, human IL-1β (117–269), TAB2-NZF domain (668–691 aa), and PAP80-UIM were inserted into plasmids pGEX-4T-1. cDNAs encoding CC domain (296–400 aa) and its mutant M3 were inserted into a modified pGEX-4T-1 vector in which the GST tag was replaced with a MBP tag. cDNAs encoding ubiquitin and its mutants were inserted into a modified pET-14b vector in which no tag was expressed. His6-tagged E1 and TRAF6 were expressed in Sf9 cells. The E. coli strain BL21(DE3/pLys) harboring plasmids of Ubc13 was induced with 1 mM IPTG at 37 °C for 4 h, and all the other plasmids listed above were induced with 0.1 mM IPTG at 16 °C overnight. Ubiquitin and ubiquitin mutants without tags were expressed in E. coli. The cells were lysed in hypotonic buffer and proteins were released by sonication. Proteins were further purified by sequential HiTrap Q column, HiTrap SP column, and size-exclusion column Superdex 75. Fractions from Superdex 75 containing ubiquitin were pooled, buffer-exchanged to H₂O, and stored at −80 °C. His-tagged proteins and GST-tagged proteins were purified using Nickel-agarose beads and glutathione sepharose beads, respectively, according to the manufacturers' instruction. MBP-tagged proteins were purified using amylose resin, according to the manufacturer's instruction. Flag-tagged TRAF6 and its mutants were transiently expressed in HEK293T cells for 36 h and purified by using anti-Flag M2 magnetic beads (Sigma), according to the manufacturer's instructions.

**Cell culture and transfection**. MEFs, HEK293T, and HeLa cells were cultured in Dulbecco's Modified Eagle's Medium supplemented with 10% (v/v) fetal bovine serum (Gibco), penicillin (100 U ml⁻¹), and streptomycin (100 mg ml⁻¹). Jurkat T cells were cultured in RPMI 1640 supplemented with 10% (v/v) fetal bovine serum (Gibco), 2 mM β-mercaptoethanol, and antibiotics. HEK293T cells were transfected with Lipofectamine 2000 (Invitrogen) or polyethyleneimine (Mr 40,000; Polysciences), according to the manufacturers' instructions.

**Retrovirus production and the generation of stable cell lines**. For production of retroviral supernatants, HEK293T cells were co-transfected with the indicated retroviral vector together with pCL-Ampho using polyethyleneimine. Four hours after transfection, media were removed and fresh media were added. Virus stocks were prepared by collecting the media 48 h after transfection. MEF cells were infected with virus in the presence of Polybrene (1 μg ml⁻¹). Twenty-four hours after infection, the media were changed to fresh media containing 1 μg ml⁻¹ puromycin. Puromycin-resistant cell pools were used for further experiments.

**Purification and fractionation of K63 polyubiquitin chains**. Polyubiquitin chains were synthesized as "In vitro ubiquitination assay" except the reaction volume was scaled up to 1 ml and concentration of each component (E1, Ubc13/Uev2, TRAF6) was doubled. After incubation, the reaction products were quenched by 20 mM

NEM at room temperature for 15 min, and then DTT was added to a final concentration of 10 mM. Following incubation at room temperature for 15 min, the final products were mixed with 0.5 ml of Nickel beads and incubated for 2 h at 4 °C to remove His6-tagged E1, Ubc13/Uev2, and TRAF6. The supernatant was separated from the beads by centrifugation and concentrated to about 500 μl before applied to size-exclusion column Superdex 200. Fractions containing different lengths of polyubiquitin chains were collected and stored at −80 °C for TAK1 activation assay.

**Pull down and detection of K63 polyubiquitin chains**. The K63-linked polyubiquitin chains were pulled down by using GST-TAB2-UBD-based method as being described[50]. Briefly, to capture K63-linked polyubiquitin chains from in vitro IKK activation reaction mixtures, the reaction was stopped by adding 20 mM NEM and incubated for 5 min at room temperature. The products were then incubated with recombinant GST-TAB2-UBD protein (2 μg each sample) and glutathione sepharose beads in the presence of lysis buffer B (50 mM Tris-HCl, pH 7.5, 500 mM NaCl, 20 mM NEM, 1 mM sodium orthovanadate, 5 mM NaF, 5 mM β-Glycerophosphate, 1 mM EDTA, 1 mM EGTA, 10 μg ml⁻¹ leupeptin, 1 mM PMSF, and 1% Triton X-100). After incubation at 4 °C for 4 h, the beads were then washed five times with lysis buffer B and one time with buffer A. Bound proteins were separated on 8–16% SDS-PAGE and immunoblotted with K63-specific antibody and other antibodies as specified in figures.

To capture endogenous K63-linked polyubiquitin chains from IL-1β-treated MEF cells, the same procedure was carried out except GST-RAP80-UIMs were used to replace GST-TAB2-UBD[46].

**Reporter assays**. HEK293T cells were transfected in triplicate with test plasmids plus 50 ng of firefly luciferase reporter plasmids (pGL4.2–3 × κB-Luc) and 5 ng of sea pansy luciferase reporter plasmids (pTK-RL) using polyethylenimine in 24-well plates. Twenty-four hours after transfection, cells were harvested and lysed with lysis buffer C (25 mM glycylglycine, 15 mM MgSO₄, 4 mM EGTA, 0.5% CHAPS, 0.5 mM DTT, 1 mM PMSF). Cell lysates were collected for measuring firefly and sea pansy luciferase activities.

**Partial purification of endogenous TAK1 kinase complex**. To partially purify endogenous TAK1 kinase complex, 15 ml of Jurkat T S100 was applied to HiTrap Q column (connection of three 5-ml columns) with buffer Q/A (20 mM HEPES-KOH, pH 7.4, 10% Glycerol, 1 mM EDTA, 1 mM EGTA, 1 mM DTT, 1 mM PMSF), which was eluted with five column volumes of a linear gradient of 0–30% buffer Q/B (Q/A with 1 M NaCl). Distribution of TAK1 was detected by immunoblotting using TAK1 total antibody, and those fractions containing TAK1 were pooled and buffer exchanged by ultrafiltration into buffer Q/A. The mixtures were applied to a 5-ml HiTrap Heparin column with buffer Q/A and eluted with five column volumes of a linear gradient of 0–50% buffer Q/B. Fractions containing TAK1 were pooled and buffer exchanged by ultrafiltration into buffer Q/A. The pooled fractions were applied to a 1-ml HiTrap Blue column with buffer Q/A and eluted with a linear gradient of 0–100% buffer Blue/B (Q/A with 2 M NaCl) in a total of volume of 5 ml. Fractions containing TAK1 were pooled and concentrated before loading onto a 24-ml Superdex 200 column, which was pre-equilibrated with buffer Q/C (Q/A with 100 mM NaCl), and eluted with one column volume of buffer Q/C. Finally, fractions containing TAK1 were collected and stored at −80 °C for TAK1 activation assay.

**Immunoprecipitation**. Cells were left untreated or treated as indicated in the figure legends and washed one time with cold phosphate-buffered saline. Cells were lysed in buffer A (20 mM Tris-HCl, pH 7.4, 150 mM NaCl, 0.5% NP-40, 10% Glycerol, 1 mM DTT, and complete protease inhibitor cocktail) for 10 min on ice and centrifuged at 20,000 × g for 10 min. Protein concentration was measured on the clarified lysates, and equal amounts of proteins were then processed for immunoblotting, immunoprecipitation, or pull-down assays, according to standard protocols. To prevent the deubiquitination of proteins, cells were lysed in buffer A containing 10 mM NEM.

**CC domain alignment**. Amino acid residues of CC domains from different species were aligned by Clustal Omega (EMBL-EBI). Positions of the CC heptad repeats, a-b-c-d-e-f-g, are indicated. Depicted sequences are as follows: mouse TRAF6, 299–356 (NP_033450.2); human TRAF6, 288–348 (NP_665802.1); rat TRAF6, 302–356 (NP_001101224.1); bovine TRAF6, 310–368 (XP_005216441.1); pig TRAF6, 312–367 (NP_001098756.1); zebrafish TRAF6, 311–373 (NP_001038217.1).

**RNA isolation and quantitative real-time PCR (qRT-PCR)**. Total RNAs were extracted using TRIzol (Invitrogen), according to the manufacturer's instructions. Total RNA (1 μg) was reverse transcribed using PrimeScript RT reagent kit (TaKaRa). Quantitative PCR was performed using SYBR Green (TaKaRa) and the ABI 7500 real-time PCR system (Applied Biosystems). The following primers were used: mouse β-actin-F, 5′-aacagtccgcctagaagcac-3′; mouse β-actin-R, 5′-cgtttga-catccgtaaagacc-3′; mouse A20-F, 5′-gaacaatgtcccgtgtc-3′; mouse A20-R, 5′-

acctactcgttggctt-3′; mouse IL-6-F, 5′-agttgccttcttgggactga-3′; mouse IL-6-R, 5′-tccacgatttcccagagaac-3′.

**Preparation of cytosolic extracts from Jurkat T cells**. Jurkat T cells were collected from 1 L culture and resuspended in an equal volume of hypotonic buffer (20 mM HEPES-KOH, pH 7.4, 10 mM KCl, 1.5 mM MgCl₂, 1 mM EDTA, 1 mM EGTA, 1 mM DTT, 1 mM PMSF). After incubation on ice for 15 min to allow cell swelling, the cells were homogenized using a Dounce homogenizer. Cell debris was removed by centrifugation at $20,000 \times g$ for 30 min, and the resulted supernatant (S20) was stored at $-80\,^\circ$C or undergone further ultracentrifugation at $100,000 \times g$ for 60 min. The cleared supernatant (S100) was collected and stored at $-80\,^\circ$C for future in vitro IKK activation assays.

**In vitro ubiquitination assay**. The basic ubiquitination assay was performed in a reaction volume of 10–20 µl containing recombinant E1 (80 nM), Ubc13/Uev2 (1 µM), ubiquitin (50 µM), and TRAF6 (20 nM) or its mutants (20 nM) in ATP buffer. The mixture was incubated at 30 °C for 30 min and stopped by addition of denaturing sample buffer followed by 95 °C treatment for 5 min. The samples were then resolved on 6–18% SDS-PAGE and analyzed by immunoblotting with an anti-ubiquitin antibody. Variation from this basic ubiquitination assay was performed as specified in the corresponding figures.

**Preparation of ubiquitin charged GST-Ubc13**. GST-Ubc13 (C87S) was incubated with 500 nM E1 and a 2-fold molar excess of ubiquitin in 50 mM Tris-HCl, pH 7.5, 2 mM ATP, 5 mM MgCl₂, and 0.1 mM DTT at 37 °C for 3 h. The reaction mixtures were separated by Superdex 75 and the purified GST-Ubc13 (C87S)~Ub was collected and stored at $-80\,^\circ$C for further analysis.

**In vitro TAK1 kinase activation assay by polyUb chains**. Partially purified TAK1 kinase complex (~1 nM) were incubated with polyUb chains, Ubal (1 µM), ATP reaction buffer, and okadaic acid (0.1 µM) in a total volume of 10–20 µl at 30 °C for 30 min. The reaction products were separated on 10% SDS-PAGE. TAK1 activation was determined by immunoblotting using TAK1 phospho-Thr187-specific antibody and using TAK1 total antibody to detect its slow mobility shift.

**GST-Ubc13 pull-down assay**. Purified GST-Ubc13 (or GST alone as negative controls) proteins were mixed with TRAF6 or its mutants in lysis buffer A (20 mM Tris-HCl, pH 7.4, 150 mM NaCl, 0.5% NP-40, 10% glycerol) together with glutathione sepharose beads. After incubation at 4 °C for 60 min, the beads were washed three times with lysis buffer A and the bound proteins were analyzed by immunoblotting with antibodies as indicated in the respective figures.

**Rate determination of polyUb chain synthesis by TRAF6**. After ubiquitination reaction as described in "In vitro ubiquitination assay", polyUb chain products were separated by 6–16% SDS-PAGE and detected by anti-Ub immunoblotting. Intensities of polyUb signals with sizes above Ub5 were quantified using ImageJ. The resulting numbers were then divided by time to get rates $R_{ob}$. The rates are relative and have an arbitrary unit.

**Determination of Km for Ub-Ubc13**. Ubc13/Uev2 (5 µM) were incubated with E1 (500 nM) and Ub (50 µM) in 50 mM Tris-HCl, pH 7.5, 2 mM ATP, 5 mM MgCl₂, and 0.1 mM DTT at 30 °C for 10 min. After desalting by G-25 (GE Healthcare) to remove ATP, 5 µl of the Ub~Ubc13/Uev2 intermediate was resolved by non-reducing SDS-PAGE and detected by Coomassie blue staining to quantify the Ub~Ubc13/Uev2 concentration, according to standard BSA proteins. Indicated amounts of Ub~Ubc13/Uev2 were incubated with T6RZC or T6RZ at 30 °C for 15 min. Reaction products were resolved by SDS-PAGE, immunoblotted with anti-Ub antibody, and quantified by Image J (NIH). Estimation of Km was performed by fitting at least duplicate measurements of data points to the Michaelis–Menten equation using GraphPad Prism.

**Analysis of TRAF6 oligomerization using native gels**. Plasmids encoding MBP-TRAF6-Flag and MBP-TRAF6-M3-Flag were transfected into HEK293T/TRAF6 KO cells. Twenty-four hours later, TRAF6 and the mutant were purified by using anti-Flag M2 magnetic beads. The eluted proteins were resolved by 10% native gel and immunoblotted with Flag antibody.

**In vitro cell-free IKK activation assay by TRAF6**. TRAF6 proteins or its variants were added into the Jurkat T S100 (10–15 µg total proteins in 10 µL reaction volume) in the presence of 0.1 µM okadaic acid and ATP buffer (50 mM Tris-HCl, pH 7.5, 2 mM ATP, 5 mM MgCl₂, 0.5 mM DTT) and incubated at 30 °C for 60 min. The reaction mixtures were separated on 10% SDS-PAGE. Activation of IKK was determined by immunoblotting using anti-IκBα phospho-Ser42/46-specific antibody and total IκBα antibody to detect its slow mobility shift after phosphorylation.

**In vitro TAK1 kinase activation assay by TRAF6**. Partially purified TAK1 kinase complex (~1 nM) were incubated with E1 (10 nM), Ubc13/Uev2 (0.5 µM), Ub (50 µM), Ubal (1 µM), ATP reaction buffer, okadaic acid (0.1 µM), and TRAF6 (or the mutants as specified in figures, 20 nM) in a total volume of 10–20 µl at 30 °C for 30 min. The reaction products were separated on 10% SDS-PAGE. TAK1 activation was determined by immunoblotting using TAK1 phospho-Thr187-specific antibody and using TAK1 total antibody to detect its slow mobility shift.

**Data availability**. All relevant data are available from the corresponding author upon reasonable request.

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

## Acknowledgements

We thank Dr Hui Xiao for the TRAF6 KO MEF cells and Dr Wei Chen, Wei Liu, and Long Zhang for critical reading of the manuscript. This study was supported by the National Basic Research Program of China (973 Program, 2013CB945000 to Z.X.), the National Key Research and Development Program of China, Stem Cell and Translational Research (2016YFA0100300 to Z.X.), the Natural Science Foundation of China (31371416 and 31571445 to Z.X., 31401200 to X.H.), The Natural Science Foundation of Zhejiang Province, China (R2110588 to Z.X.), the Key Program of Zhejiang Provincial Natural Science Foundation of China (LZ16C050001 to D.N.).

## Author contributions

L.H. and Z.X. conceived and designed the study, analyzed all the data, and co-wrote the manuscript. L.H. and J.X. performed all the experiments. X.X., Y.Z., P.T., H.L., X.H. and C.W. helped with protein purification and with cell assays. J.L. and P.X. contributed reagents and edited the manuscript. D.N. helped with cellular quantification of Ubc13 and edited the manuscript.

## Additional information

**Competing interests:** The authors declare no competing financial interests.

