## [Peer Review File · Nature Communications]

Reviewers' comments:

Reviewer #1 (Remarks to the Author):

General comments:

The manuscript by Hu et al describes a detailed functional examination of the various domains of TRAF6 in supporting unanchored, processive, lysine (K63-linked) polyubiquitin chain synthesis. Studies were conducted by dominant-positive ectopic expression as well as E2 conjugase interaction with the TRAF6 coiled-coil (CC) domain and activity rescue in of IL-1 signaling in TRAF6 knock-out cells that is lost with specific CC mutants.

These studies are especially significant because they repeat and extend earlier studies focusing on TRAF6-anchored K63-linked auto-polyubiquitin chain synthesis and NF- κ B activity. One of these earlier studies reported that the TRAF6 Ring Finger in the presence of at least one Zn Finger domain is capable of NF- κ B activation in the absence of the (coiled-coil) CC domain, when artificially multimerized [Ref 20_Baud et al., 1999]. Another report demonstrated that the amino Ring-Zn-CC is capable of ubiquitination and NF- κ B activation in the absence of the TRAF-C domain, while the TRAF-C is inhibitory in the absence of the Ring-Zn [Wang et al., 2010]. Consequently, the specific role of the CC domain, independent of processivity, as being either simply a passive TRAF oligomerization structure for supporting RZ function or an active component in polyubiquitination has never been resolved, although previous reports have suggested that the CC is involved in E2 conjugase recruitment [Ref 23_Yang et al., 2004; Yin et al., 2009; Ref 22_Wang et al., 2010]. The manuscript by Hu et al now provides compelling evidence for Ubc13/Uev1A E2 conjugase interaction with the CC and its importance for processivity.

Of particular note, the rescue experiments described in association with Figures 7 and S6 nicely demonstrate the importance of the CC domain in rescuing IL-1 receptor signaling in TRAF6 KO MEF cells. It would be interesting to know whether the use of knock-out cells was required for this approach to work. This is because it was previously reported [Ref22_Wang et al., 2010] that TRAF6 ubiquitination is mediated in trans via the CC domain. Interestingly, this implies that the experiment might not have revealed any difference if conducted in the 293T cells used for the other studies in the manuscript.

The deletion results presented by Hu et al also demonstrate an amazing similarity between unanchored K63 polyubiquitin chain synthesis and that previously reported for TRAF6-anchored synthesis and corresponding NF- κ B activity [Ref22_Wang et al., 2010]. This similarity, which was not noted by Hu et al, additionally supports an argument that both TRAF6-anchored and unanchored K63 poly Ub chains may be the result of a single mechanism, further complicating an understanding of the mechanism by which unanchored K63-linked poly Ub is synthesized. For example, is it derived by anchored chain "shedding" or by direct synthesis? This remains a challenge because of the activity of non-ubiquitinated TRAF6 [Walsh et al., 2008], the apparent ability of TRAF6 to generate both types of chains, depending upon the E2 conjugase [Windheim et al., 2008], and the possibility that the 26S proteasome could be involved in unanchored K63-linked poly Ub shedding [Nanduri, et al., 2015].

References not found in manuscript:

Yin et al., 2009: Yin, Q., Lin, S-C., Lamothe, B., Lu, M., Lo, Y-C., Hura, G., Zheng, L., Rich, RL., Campos, AD., Myszka, DG., Lenardo, MJ., Darnay, BG., & Wu, H. E2 interaction and dimerization in the crystal structure of TRAF6. *Nature Structural & Molecular Biology* 16, 658-666, doi:10.1038/nsmb.1605 (2009).

Walsh et al., 2008: Walsh, MC, Ki, GK., Maurizio, PL, Molnar, EE, & Choi, Y. TRAF6 Autoubiquitination-independent activation of the NF κ B and MAPK pathways in response to IL-1 and

RANKL. *PLoS One* 3, doi: 10.1371/journal.pone.0004064 (2008).

Nanduri, et al., 2015: Nanduri, P, Hao, R, Fitzpatrick, T & Yao, T-P. Chaperone-mediated 26S proteasome remodeling facilitates free K63 ubiquitin chain production and aggresome clearance. *J. Biol. Chem.* 290, 9455-9464, doi: 10.1074/jbc.M114.627950 (2015).

References in manuscript:

Ref 20_Baud et al., 1999

Ref 22_Wang et al., 2010

Ref 23_Yang et al., 2004

Issues of specific note:

- Page 2, Line 30: The sentence "Fusion of the CC domain to the E3 CHIP ..." is somewhat unclear and "to the E3 CHIP" might be replaced by "to the E3 ubiquitin ligase CHIP/STUB1".
- Page 8, Line 151: Fraction B7 was stated as having "almost no activities". However, the data of Figure 1E reveals what appears to be weak, but significant activity. Perhaps the previous statement could be replaced with "very weak activity".
- Page 10, Lines 191-192: The NF- κ B activity reduction for the CC is virtually identical to that previously reported in reference 22, which should be cited.
- Pages 19-20, Lines 393-397: The conclusion that E2 conjugases "possesses a higher order structure" determined from "data not shown" for N-terminal Flag-tagged Ub vs. non-tagged being deficient in TAK1 activation may not be unique. Why couldn't this simply be the result of a requirement for a free N-terminus in TAK1 binding, which is occluded by the Flag tag?
- Page 22, Lines 459-462 and Figure 7C: The summary sentence, along with the cartoon presented in Figure 7C, imply that CC domain "oligomerization" via the coiled-coil CC domain is of a higher order than the ternary associations observed in structural reports for this domain in TRAF family proteins. The authors appear to be using the term "oligomerization" in a generalized and confusing fashion that equates the trimeric structures associated with TRAF family molecules with a higher-order association. To this reviewer's knowledge, the observed stoichiometry for CC domains in TRAF molecules is always toward trimers, i.e., 3-chains (McWhirter, SM, et al., *PNAS* 96:8408, 1999; Ye, H., et al., *Nature* 418:443, 2002; Ni, C-Z, et al., *J. Immunol.* 173:7394, 2004; Park, YC, *Nature* 398:533, 1999). This is not to imply that higher order structures have not been theorized and reported for parallel, anti-parallel, and mixed parallelism in coiled-coils (e.g., Parry, DAD, et al., *J. Struct. Biol.* 163:258, 2008; Bromley, EHC, et al., *HFSP J.* 3:201, 2009). TRAF may be limited to trimeric associations because of the presence of the bulky C-TRAF domain, that also supports TRAF oligomerization, providing both an additional source of interactions as well as an impediment to the formation of higher-order structures. Consequently, such higher-order structures may require unique TRAF conformational native and denatured states, for which there is presently an absence of evidence. A more likely scenario could involve the "Ub-Ub non-covalent interactions between polyUb chains" suggested by the authors (Page 20, Lines 402-403). Overall, the authors should distinguish between classic TRAF oligomerization (likely trimeric) and higher-order associations.
- Page 32: Panels C and E list "M.W.", but should also state that the units are "kDa". Figures 2, 3, 4, and 5 on Pages 34, 36, 38 are similarly unlabeled. It is recommended that simply describing this in the figure legend for Figure 1 could remedy this issue.
- Page 33, Line 678: Figure 1 legend for panel "E" should indicate that the S.E. and L.E. labels in the figure refer, respectively, to "short exposure" and "long exposure."

Reviewer #2 (Remarks to the Author):

TRAF6 is a ubiquitin E3 ligase that functions together with the Ubc13/Uev1A ubiquitin E2 complex to catalyze the synthesis of K63 polyubiquitin chains, which activate TAK1 and IKK. TRAF6 contains a RING domain that is important for interacting with Ubc13 to synthesize polyubiquitin chains. TRAF6 also contains a C-terminal coiled-coil (CC) domain that is important for TAK1 and IKK activation, but the mechanism by which the CC domain mediates kinase activation is not well understood. In this paper, Xia and colleagues showed that the TRAF6 CC domain is important for the processive synthesis of long polyubiquitin chains that activate TAK1. Using elegant in vitro reconstitution assays, they showed that the TRAF6 CC domain not only mediates oligomerization but also interacts directly with Ubc13. This finding may explain at least in part why TRAF6 stimulates K63 polyubiquitin chain synthesis in a highly processive manner. The authors also confirmed their findings in cell-based assays. Overall, the quality of the data is high and the findings are important. However, the paper can be improved by addressing the following questions.

- 1) Figure 4C shows that T6RZ can also synthesize long polyubiquitin chains, similarly to those synthesized by T6RZC (compare lane 6 and 13). It seems that given enough time, long ubiquitin chains can be synthesized by T6RZ. Do these ubiquitin chains activate TAK1? If not, why are these long ubiquitin chains incapable of activating TAK1?
- 2) Figure 3F shows that CHIP-Z3CC but not CHIP can activate NF- κ B. Is fusion of TRAF6 CC domain (no ZF domains) to CHIP sufficient to activate NF- κ B? What's difference between the ubiquitin chains synthesized by CHIP-Z3CC and CHIP?
- 3) The authors discussed extensively how long polyubiquitin chains activate TAK1 but the proposed models are quite speculative and not based on evidence. The discussion needs to be revised.

Responses to Reviewers' Comments.

Reviewer #1 (Remarks to the Author):

General comments:

The manuscript by Hu et al describes a detailed functional examination of the various domains of TRAF6 in supporting unanchored, processive, lysine (K63-linked) polyubiquitin chain synthesis. Studies were conducted by dominant-positive ectopic expression as well as E2 conjugase interaction with the TRAF6 coiled-coil (CC) domain and activity rescue in of IL-1 signaling in TRAF6 knock-out cells that is lost with specific CC mutants.

These studies are especially significant because they repeat and extend earlier studies focusing on TRAF6-anchored K63-linked auto-polyubiquitin chain synthesis and NF- κ B activity. One of these earlier studies reported that the TRAF6 Ring Finger in the presence of at least one Zn Finger domain is capable of NF- κ B activation in the absence of the (coiled-coil) CC domain, when artificially multimerized [Ref 20_Baud et al., 1999]. Another report demonstrated that the amino Ring-Zn-CC is capable of ubiquitination and NF- κ B activation in the absence of the TRAF-C domain, while the TRAF-C is inhibitory in the absence of the Ring-Zn [Wang et al., 2010]. Consequently, the specific role of the CC domain, independent of processivity, as being either simply a passive TRAF oligomerization structure for supporting RZ function or an active component in polyubiquitination has never been resolved, although previous reports have suggested that the CC is involved in E2 conjugase recruitment [Ref 23_Yang et al., 2004; Yin et al., 2009; Ref 22_Wang et al., 2010]. The manuscript by Hu et al now provides compelling evidence for Ubc13/Uev1A E2 conjugase interaction with the CC and its importance for processivity.

We thank the reviewer for his/her enthusiasm about our manuscript. Our initial intention was to define the domain(s) that are responsible for the processivity of TRAF6 E3 ubiquitin ligase activity. We eventually narrowed this down to the CC domain and provided experimental evidences to show that the CC domain mediates TRAF6 oligomerization and interaction with Ubc13/Uev1A (or 2) therefore conferring processivity to TRAF6 E3 ubiquitin ligase activity. Our results linked the structural element CC domain of TRAF6 to its biochemical function.

Of particular note, the rescue experiments described in association with Figures 7 and S6 nicely demonstrate the importance of the CC domain in rescuing IL-1 receptor signaling in TRAF6 KO MEF cells. It would be interesting to know whether the use of knock-out cells was required for this approach to work. This is because it was previously reported [Ref22_Wang et al., 2010] that TRAF6 ubiquitination is mediated in trans via the CC domain. Interestingly, this implies that the experiment might not have revealed any difference if conducted in the 293T cells used for the other studies in the manuscript.

For this question, we reasoned it depends on whether mutants M3, M4, and M5 would function as dominant-negative (DN) mutants or not in the presence of TRAF6 WT. If they were dominant-negative mutants, then they would still show defects in IL-1R signaling even in the presence of TRAF6 WT such as in wild-type cells.

We tested this by expressing WT, M3, M4 or M5 in C6 (a 293T cell line stably expressing IL-1R) and checking their response to IL-1 β stimulation. As shown below, in agreement with what we obtained in TRAF6 KO MEF cells, cells expressing M3, M4, or M5 still showed defects in TAK1 activation (The level of p-TAK1 was much reduced in M3-, M4- or M5-expressing cells when compared to what in TRAF6 WT-expressing cells.). Our explanation is as follows. For M3, although it is defective in forming complex with TRAF6 WT, it can still interact with IRAK1 (through TRAF-C domain, the upstream intermediate of the IL-1R signaling pathway), and therefore would function as a dominant-negative mutant, especially when its concentration is much higher than WT. For M4 or M5, on one hand, they would function like M3 as DN mutants by binding with IRAK-1 if they form complex by themselves. On the other hand, they can still form complex with TRAF6 WT. But this kind of M4- or M5-containing TRAF6 complex, due to compromised interaction with Ubc13, would be comprised in its processivity when catalyzing polyUb chain assembly leading to compromised IL-1R signaling.

The deletion results presented by Hu et al also demonstrate an amazing similarity between unanchored K63 polyubiquitin chain synthesis and that previously reported for TRAF6-anchored synthesis and corresponding NF- κ B activity [Ref22_Wang et al., 2010]. This similarity, which was not noted by Hu et al, additionally supports an argument that both TRAF6-anchored and unanchored K63 poly Ub chains may be the result of a single mechanism, further complicating an understanding of the mechanism by which unanchored K63-linked poly Ub is synthesized. For example, is it derived by anchored chain “shedding” or by direct synthesis? This remains a challenge because of the activity of non-ubiquitinated TRAF6 [Walsh et al., 2008], the apparent ability of TRAF6 to generate both types of chains, depending upon the E2 conjugase [Windheim et al., 2008], and the possibility that the 26S proteasome could be involved in unanchored K63-linked poly Ub shedding [Nanduri, et al., 2015].

We are sorry for this carelessness. We have now cited this paper, Wang et al. in the main text.

In terms of the mechanisms of unanchored K63 polyUb chain synthesis, it is interesting to discuss a little bit more here. Because of its own reactivity towards lysine 63 on ubiquitin presented by Uev1 instead of substrate lysine residues, Ubc13/Uev1 cannot conjugate ubiquitin (or ubiquitin chains) directly to a substrate, which needs another E2 to “help” add the first Ub to a substrate (Petroski et al., 2007, Windheim et al., 2008, Stewart et al., 2016). In the case of TRAF6-Ubc13/Uev1, unanchored K63 polyUb chains were assembled directly when using recombinant proteins *in vitro* in our hands, as was also reported by Petroski et al., 2007 and Windheim et al., 2008. In the case of TRAF6-UbcH5c, both unanchored and conjugated polyUb chains (to TRAF6) were produced directly when using recombinant proteins *in vitro* in our hands and in Windheim et al., 2008. But in this case, the linkages were mixed. In addition, genetic studies showed that only Ubc13

but not any other E2s have been reported to be critical in IL-1R signaling, suggesting in cells any putative priming E2s for TRAF6 autoubiquitination might not be essential for IL-1R signaling (Fukushima et al., 2007; Yamamoto et al., 2006). Therefore, based on current knowledge, we favor the possibility that unanchored polyUb chains by TRAF6-Ubc13/Uev1 can be synthesized directly both *in vitro* and *in vivo*. But we agree with this reviewer that in general it is highly possible that both modes could work: direct synthesis or “shedding” from anchored polyUb chains by deubiquitinating enzymes (DUBs) with “en bloc” DUB activity like Poh1. It is also possible both modes could co-exist.

Petroski, M. D., et al. (2007). "Substrate modification with lysine 63-linked ubiquitin chains through the UBC13-UEV1A ubiquitin-conjugating enzyme." J Biol Chem **282**(41): 29936-29945.

Windheim, M., et al. (2008). "Two different classes of E2 ubiquitin-conjugating enzymes are required for the mono-ubiquitination of proteins and elongation by polyubiquitin chains with a specific topology." Biochem J **409**(3): 723-729.

Stewart, M. D., et al. (2016). "E2 enzymes: more than just middle men." Cell Res **26**(4): 423-440.

Yamamoto, M., et al. (2006). "Key function for the Ubc13 E2 ubiquitin-conjugating enzyme in immune receptor signaling." Nat Immunol **7**(9): 962-970.

Fukushima, T., et al. (2007). "Ubiquitin-conjugating enzyme Ubc13 is a critical component of TNF receptor-associated factor (TRAF)-mediated inflammatory responses." Proc Natl Acad Sci U S A **104**(15): 6371-6376.

References not found in manuscript:

Yin et al., 2009: Yin, Q., Lin, S-C., Lamothe, B., Lu, M., Lo, Y-C., Hura, G., Zheng, L., Rich, RL., Campos, AD., Myszka, DG., Lenardo, MJ., Darnay, BG., & Wu, H. E2 interaction and dimerization in the crystal structure of TRAF6. Nature Structural & Molecular Biology 16, 658-666, doi:10.1038/nsmb.1605 (2009).

I guess the reviewer was to mean Windheim et al., 2008. We have now cited this paper in the main text. (The paper above is Ref 17 in our original version and now Ref 21 in this revised version.)

Walsh et al., 2008: Walsh, MC, Ki, GK., Maurizio, PL, Molnar, EE, & Choi, Y. TRAF6 Autoubiquitination-independent activation of the NFkB and MAPK pathways in response to IL-1 and RANKL. PloS One 3, doi: 10.1371/journal.pone.0004064 (2008).

Nanduri, et al., 2015: Nanduri, P, Hao, R, Fitzpatrick, T & Yao, T-P. Chaperone-mediated 26S proteasome remodeling facilitates free K63 ubiquitin chain production and aggresome clearance. J. Biol. Chem. 290, 9455-9464, doi: 10.1074/jbc.M114.627950 (2015).

References in manuscript:

Ref 20_Baud et al., 1999
Ref 22_Wang et al., 2010
Ref 23_Yang et al., 2004

Thanks for pointing out these papers. We have now cited them in the revised main text.

Issues of specific note:

- Page 2, Line 30: The sentence “Fusion of the CC domain to the E3 CHIP ...” is somewhat unclear and “to the E3 CHIP” might be replaced by “to the E3 ubiquitin ligase CHIP/STUB1”.

We apologize for this vagueness. We have now changed it in this revised version according to your suggestion.

- Page 8, Line 151: Fraction B7 was stated as having “almost no activities”. However, the data of Figure 1E reveals what appears to be weak, but significant activity. Perhaps the previous statement could be replaced with “very weak activity”.

Thanks for pointing this out. We have now changed it in this revised main text.

- Page 10, Lines 191-192: The NF- κ B activity reduction for the CC is virtually identical to that previously reported in reference 22, which should be cited.

We thank the reviewer for pointing this out. We have modified our wording to cite this paper.

- Pages 19-20, Lines 393-397: The conclusion that E2 conjugases “possesses a higher order structure” determined from “data not shown” for N-terminal Flag-tagged Ub vs. non-tagged being deficient in TAK1 activation may not be unique. Why couldn’t this simply be the result of a requirement for a free N-terminus in TAK1 binding, which is occluded by the Flag tag?

What we observed was that N-terminal tagged or modified (by fluorescein) ubiquitin is comparable to unmodified ubiquitin in polyUb chain assembly catalyzed by TRAF6 and Ubc13. But when we purified these chains and used them in TAK1 activation assay, they did not show detectable TAK1-stimulatory activities. There could be several possibilities to explain it. One, as we mentioned in our Discussion section, is that the unmodified N-terminus is essential for higher-order structure formation. Another possibility is, as you suggested, it might require a free unmodified original N-terminus for TAK1 binding.

- Page 22, Lines 459-462 and Figure 7C: The summary sentence, along with the cartoon presented in Figure 7C, imply that CC domain “oligomerization” via the coiled-coil CC domain is of a higher order than the ternary associations observed in structural reports for this domain in TRAF family proteins. The authors appear to be using the term “oligomerization” in a generalized and confusing fashion that equates the trimeric structures associated with TRAF family molecules with a higher-order association. To this reviewer’s knowledge, the observed stoichiometry for CC domains in TRAF molecules is always toward trimers, i.e., 3-chains (McWhirter, SM, et al., PNAS 96:8408, 1999; Ye, H., et al., Nature 418:443, 2002; Ni, C-Z, et al., J. Immunol. 173:7394, 2004; Park, YC, Nature 398:533, 1999). This is not to imply that higher order structures have not been theorized and reported for parallel, anti-parallel, and mixed parallelism in coiled-coils (e.g., Parry, DAD, et al., J. Struct. Biol.

163:258, 2008; Bromley, EHC, et al., HFSP J. 3:201, 2009). TRAF may be limited to trimeric associations because of the presence of the bulky C-TRAF domain, that also supports TRAF oligomerization, providing both an additional source of interactions as well as an impediment to the formation of higher-order structures. Consequently, such higher-order structures may require unique TRAF conformational native and denatured states, for which there is presently an absence of evidence. A more likely scenario could involve the “Ub-Ub non-covalent interactions between polyUb chains” suggested by the authors (Page 20, Lines 402-403). Overall, the authors should distinguish between classic TRAF oligomerization (likely trimeric) and higher-order associations.

We apologize for the insufficient explanation of the cartoon and potential confusion. We also agree with the reviewer’s comments. We were to propose that upon trimeric TRAF6 complex formation, higher-order, say dimer of trimers, TRAF6 complex would follow, which might be more beneficial for its processivity. But, as this reviewer pointed out, the experimental data are not strong at present, we therefore have modified our model to stress the formation of trimeric complex without implicating higher-order structure formation. Accordingly, we have also modified our wording in the figure legend.

- Page 32: Panels C and E list “M.W.”, but should also state that the units are “kDa”. Figures 2, 3, 4, and 5 on Pages 34, 36, 38 are similarly unlabeled. It is recommended that simply describing this in the figure legend for Figure 1 could remedy this issue.

Thanks for pointing this out. We have now included the new information in our revised figures and figure legends.

- Page 33, Line 678: Figure 1 legend for panel “E” should indicate that the S.E. and L.E. labels in the figure refer, respectively, to “short exposure” and “long exposure.”

Thanks for pointing this out. We have now included this information in this revised figure legend.

Reviewer #2 (Remarks to the Author):

TRAF6 is a ubiquitin E3 ligase that functions together with the Ubc13/Uev1A ubiquitin E2 complex to catalyze the synthesis of K63 polyubiquitin chains, which activate TAK1 and IKK. TRAF6 contains a RING domain that is important for interacting with Ubc13 to synthesize polyubiquitin chains. TRAF6 also contains a C-terminal coiled-coil (CC) domain that is important for TAK1 and IKK activation, but the mechanism by which the CC domain mediates kinase activation is not well understood. In this paper, Xia and colleagues showed that the TRAF6 CC domain is important for the processive synthesis of long polyubiquitin chains that activate TAK1. Using elegant in vitro reconstitution assays, they showed that the TRAF6 CC domain not only mediates oligomerization but also interacts directly with Ubc13. This finding may explain at least in part why TRAF6 stimulates K63 polyubiquitin chain synthesis in a highly processive manner. The authors also confirmed their findings in cell-based assays.

Overall, the quality of the data is high and the findings are important. However, the paper can be improved by addressing the following questions.

We thank the reviewer for his/her positive opinion about our study and constructive suggestions to improve this study. Our specific responses to the points raised are as follows.

1) Figure 4C shows that T6RZ can also synthesize long polyubiquitin chains, similarly to those synthesized by T6RZC (compare lane 6 and 13). It seems that given enough time, long ubiquitin chains can be synthesized by T6RZ. Do these ubiquitin chains activate TAK1? If not, why are these long ubiquitin chains incapable of activating TAK1?

We agree with the reviewer. In purified system, T6RZ was able to assemble long polyubiquitin chains when given long incubation time and sufficiently high concentration. These chains were able to activate highly purified TAK1 complex (Figure below, left panel). However, T6RZ was not able to activate TAK1 in crude S20, even if T6RZ was used at much higher concentrations than T6RZC (Figure below, right panel; Figure 2c). These data imply that, as we stressed in the main text, proper processivity conferred by CC domain is critical to ensure timely accumulation of long polyubiquitin chains for TAK1 activation.

2) Figure 3F shows that CHIP-Z3CC but not CHIP can activate NF- κ B. Is fusion of TRAF6 CC domain (no ZF domains) to CHIP sufficient to activate NF- κ B? What's difference between the ubiquitin chains synthesized by CHIP-Z3CC and CHIP?

Thanks for the suggestions. We have tested this and found that CHIP-CC did not give significant NF- κ B activation in the 3 κ B-Luc reporter assay, but addition of ZnF3-ZnF4 making it active (Figure below). Our data, as shown in Figures 2 and 3, didn't suggest any functions of ZnF3-ZnF4 in TRAF6 processivity in a DIRECT way; but our data in supplementary Fig. 2b (DUB inhibitor ubiquitin aldehyde (Ubal) was able to rescue TAK1-IKK activation by TRAF6 Δ ZnF3 and Δ ZnF4) suggested that ZnF3-ZnF4 could contribute to polyUb chain synthesis and TAK1-IKK activation in an indirect way, possibly by counteracting deubiquitinating activities. It is conceivable that here in the case of CHIP-Z3CC fusion, inclusion of ZnF3-ZnF4 could also help CHIP-Z3CC in NF- κ B activation in a similar way.

Regarding the polyubiquitin chains synthesized by CHIP and CHIP-Z3CC, both of them can assemble K63-linked polyubiquitin chains, but CHIP-Z3CC can assemble the chains in a much faster rate under the same reaction condition (Figure below). We believe fusion of CC confers processivity to CHIP, making it capable of assembling and accumulating long polyUb chains for TAK1 activation, whereas CHIP alone lacks proper processivity to assemble/accumulate TAK1-stimulatory K63 polyUb chains.

3) The authors discussed extensively how long polyubiquitin chains activate TAK1 but the proposed models are quite speculative and not based on evidence. The discussion needs to be revised.

We thank the reviewer for his/her comment. We have now revised this part of the Discussion and removed our discussion on long K63 polyUb chains being “a dynamic temporary organelle”.

Our initial intention to propose that long K63 polyUb chains might adopt a higher-order structure and form a dynamic temporary organelle was to stimulate the ubiquitin field to do more research on this interesting topic. But we agree with the reviewer that currently our proposed models are somewhat speculative. Like we mentioned in the main text, advanced AFM and/or Cryo-EM imaging technologies could provide direct data on the higher-order structure of long polyUb chains (not only K63- but also other linkages).

REVIEWERS' COMMENTS:

Reviewer #1 (Remarks to the Author):

I am satisfied with the authors' responses to my previous comments.

Reviewer #2 (Remarks to the Author):

This revision has addressed most of my concerns and I am supportive of its publication

Reviewers' Comments:

Reviewer #1 (Remarks to the Author):

I am satisfied with the authors' responses to my previous comments.

Reviewer #2 (Remarks to the Author):

This revision has addressed most of my concerns and I am supportive of its publication.

We thank the two reviewers for their positive opinion of our study as well as their insightful comments to improve the quality of our study.